# TAG2TEXT: GUIDING VISION-LANGUAGE MODEL VIA IMAGE TAGGING

**Xinyu Huang**[1,2]  **Youcai Zhang**[2]  **Jinyu Ma**[2]  **Weiwei Tian**[4]  **Rui Feng**[1,4*]
**Yuejie Zhang**[1*]  **Yaqian Li**[2]  **Yandong Guo**[2]  **Lei Zhang**[3]

[1]Shanghai Key Lab of Intell. Info. Processing, School of Computer Science, Fudan University
[2]OPPO Research Institute   [3]International Digital Economy Academy (IDEA)
[4]Academy for Engineering and Technology, Fudan University

## ABSTRACT

This paper presents Tag2Text, a vision language pre-training (VLP) framework, which introduces image tagging into vision-language models to guide the learning of visual-linguistic features. In contrast to prior works which utilize object tags either manually labeled or automatically detected with an off-the-shelf detector with limited performance, our approach explicitly learns an image tagger using tags parsed from image-paired text and thus provides a strong semantic guidance to vision-language models. In this way, Tag2Text can utilize large-scale annotation-free image tags in accordance with image-text pairs, and provides more diverse tag categories beyond objects. As a result, Tag2Text demonstrates the ability of a foundational image tagging model, with superior zero-shot performance even comparable to fully supervised models. Moreover, by leveraging the tagging guidance, Tag2Text effectively enhances the performance of vision-language models on both generation-based and alignment-based tasks. Across a wide range of downstream benchmarks, Tag2Text achieves state-of-the-art results with similar model sizes and data scales, demonstrating the efficacy of the proposed tagging guidance. Codes, demo and pre-trained models are available at https://github.com/xinyu1205/recognize-anything.

## 1 INTRODUCTION

Vision language pre-training (VLP) has shown an effective approach for learning a generic multi-modal representation and improving vision-language (VL) tasks including generation-based (*e.g.,* image captioning) and alignment-based (*e.g.,* image-text retrieval). As large-scale datasets of image-text pairs (Sharma et al., 2018; Changpinyo et al., 2021; Schuhmann et al., 2021; Radford et al., 2021; Jia et al., 2021) become available, recent works mainly focus on using transformer-based models to perform contrastive (Radford et al., 2021; Jia et al., 2021; Li et al., 2021; Bao et al., 2021; Li et al., 2022; Yu et al., 2022) or generative learning (Wang et al., 2021b; Li et al., 2022; Wang et al., 2022a; Chen et al., 2022; Yu et al., 2022; Wang et al., 2022c;d; Li et al., 2023) from massive image-text pairs. While great progress has been made, such studies normally rely on brute force pre-training manners that involve direct interaction with different modality features, modeling weakly-supervised learning due to the lack of explicit alignment supervision between image and text (Li et al., 2020b; Hu et al., 2021; Zeng et al., 2021).

Prior approaches (*e.g.,* OSCAR (Li et al., 2020b), VIVO (Hu et al., 2021), X-VLM (Zeng et al., 2021)) introduce the use of object tags as anchor points to ease the learning of semantic alignments between images and texts. However, these approaches rely on obsolete detector-based VLP frameworks, which employ off-the-shelf object detectors (*e.g.,* Faster RCNN (Ren et al., 2015)) to extract image features (as shown in Figure 1 ①). The primary limitation of detector-based models is that the used object detectors are normally not perfect but have to be kept frozen during VLP to maintain detection ability, thereby restricting the capacity of vision-language models (Li et al., 2021; Dou et al., 2022; Huang et al., 2022). Moreover, utilizing object detectors leads to a substantial increase in model parameters and running time (Li et al., 2021; Kim et al., 2021). Consequently, more recent works (Li et al., 2021; 2022; Dou et al., 2022; Li et al., 2023) primarily utilize detector-free VL

---
*Corresponding author.

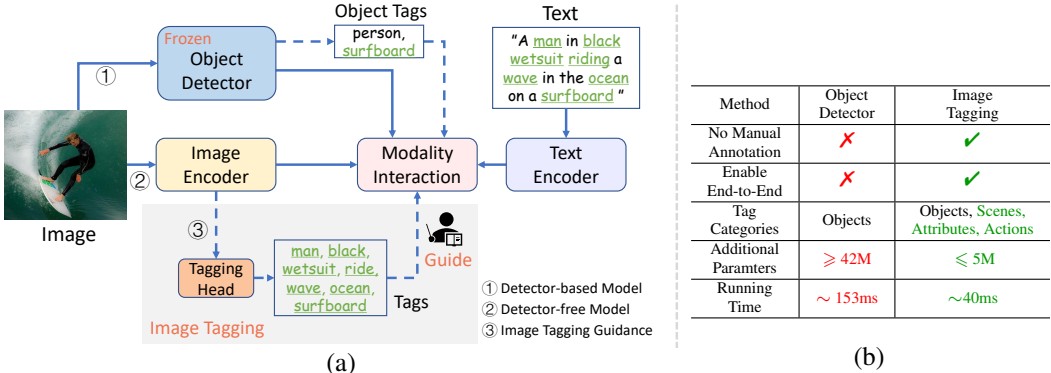

Figure 1: (a) ①: Prior works demonstrate the effectiveness of incorporating object tags into VL models based on an off-the-shelf detector. ②: Since the detector restricts the model's capacity and is time-consuming, recent VL models normally avoid using a detector, resulting in poor utilization of valuable tags. ③: We re-introduce tag guidance into detector-free VL models via image tagging with a simple tagging head. The tagging head is supervised by annotation-free image tags parsed from its paired text. Our model achieves a superior tagging ability and effectively enhances vision-language tasks. (b) Comparison of object detector and image tagging used in VL models.

models to address these limitations, resulting in the discarding of valuable tags (as shown in Figure 1 ②).

In this work, as shown in Figure 1 ③, we re-introduce tag guidance into detector-free VL models via the novel approach of image tagging. We demonstrate that integrating image tagging with other VLP tasks in a multi-task manner is a natural and effective approach from two crucial perspectives. *1)* **Data:** The pre-training image tags are obtained through automatically text semantic parsing, enabling large-scale annotation-free image tags in accordance with image-text pairs can be utilized, without the requirement of expensive grounding annotations which are necessary for object detectors. Image tagging also provides a better bridge between image and text, given that the parsed tag categories are more diverse and beyond objects, such as scenes, attributes, actions, etc. *2)* **Architecture:** Image tagging merely necessitates adding a recognition head followed by the original image encoder, ensuring efficient end-to-end pre-training and resulting in fewer parameters and improved efficiency. Figure 1(b) provides a comprehensive comparison between image tagging and object detection.

Concretely, we present Tag2Text, a VLP framework which introduces image tagging into vision-language models to guide the learning of visual-linguistic features. **For image tagging**, previous approaches primarily rely on limited manually annotated datasets (Lin et al., 2014; Everingham et al., 2015), resulting in a poor generalization capability. In contrast, Tag2Text utilizes large-scale image-text pairs, achieving an exceptional tag recognition capability of 3,429 commonly human-used categories. Remarkably, Tag2Text demonstrates a foundational image tagging capability with superior zero-shot performance, which significantly outperforms other state-of-the-art (SOTA) vision-language models such as CLIP (Radford et al., 2021), BLIP (Li et al., 2022), and BLIP-2 (Li et al., 2023) and is even comparable to fully supervised models (Ridnik et al., 2023).

Moreover, Tag2Text effectively leverages tagging guidance to enhance the performance of vision-language models. **For generation-based tasks**, we design the training task as image-tag-text generation which empowers the model to produce text descriptions based on the image features in accordance with assigned tags. As depicted in Figure 2, Tag2Text generates more comprehensive text descriptions with the guidance of comprehensively recognized tags. Additionally, Tag2Text permits users to input desired tags, providing the flexibility in composing corresponding texts (Zheng et al., 2019). **For alignment-based tasks**, while previous models rely on the alignment of multi-modal features which are considered as black-box approaches, Tag2Text augments these methods by incorporating tags as visible alignment indicators.

Our key contributions can be summarized as follows:

- For the first time, Tag2Text demonstrates the potential of a foundational image tagging model by utilizing large-scale annotation-free image tags parsed from image-paired text, exhibiting zero-shot capabilities rivalling full supervision manners.

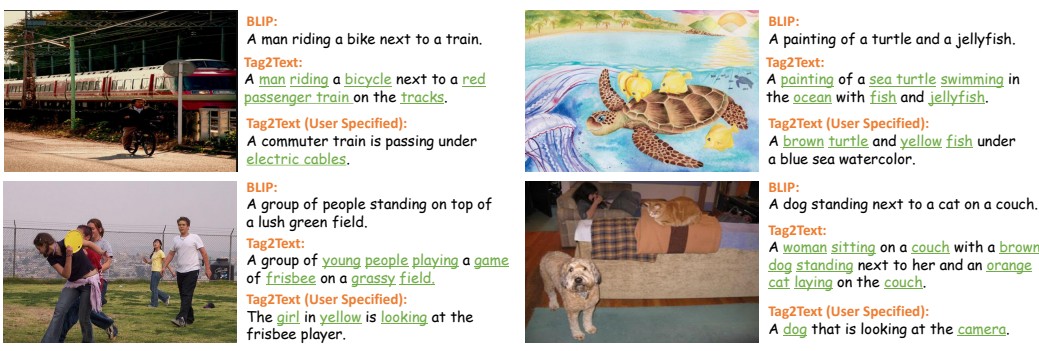

Figure 2: Comparison of image captioning results between Tag2Text (pre-training on 14M images) and BLIP (Li et al., 2022) (pre-training on 129M images). Tag2Text integrates recognized image tags as guiding elements into text generation, resulting in the generation with more comprehensive text descriptions (see Table 2 for quantitative results). Tag2Text also allows users to input specified tags to generate corresponding captions, offers a way of controlling caption generation through the use of input tags.

- Tag2Text re-introduces tag guidance into detector-free vision-language models by seamlessly integrating image tagging, effectively enhancing the performance of both generation-based tasks and alignment-based tasks.

- A wide range of downstream benchmarks, along with a series of qualitative results, demonstrate the superior tagging ability of Tag2Text and the efficacy of incorporating tagging guidance information into vision-language models.

## 2 RELATED WORK

**Vision-Language Models** consist of generation-based and alignment-based models. Generation-based models involve generating text related to an input image. The initial approach of generation-based models relies on a two-stage process of recognizing tags from an image and then using them to compose a caption (Fang et al., 2015). Notably, image features do not participate in the text generation stage. With remarkable progress of language models (Devlin et al., 2018; Brown et al., 2020; Ouyang et al., 2022), language modeling gradually becomes a dominant pre-training objective in vision-language generation-based models (Wang et al., 2021b; Li et al., 2022; Wang et al., 2022c; Chen et al., 2022; Wang et al., 2022a;d). Such an approach endows vision-language models with the capability to generate expressive captions conditioned on visual information. Distinguished from existing works, our proposed approach is a novel scheme of image-tag-text generation, enabling our model to effectively regulate the content and quality of the generated text based on assigned tags.

Alignment-based models involve determining whether an image and a text are matched. Previous models perform either image-text contrastive learning (Radford et al., 2021; Jia et al., 2021; Li et al., 2021; Bao et al., 2021; Li et al., 2022; Huang et al., 2022) with the dual-encoder architecture or image-text matching (Li et al., 2020b; 2021; Bao et al., 2021; Dou et al., 2022; Li et al., 2022) with the fusion-encoder architecture. IDEA (Huang et al., 2022) introduces identified tags as additional text supervision only enhancing image classification accuracy. These models predominantly rely on the alignment of multi-modal features which are considered as black-box approaches for retrieval task. Tag2Text augments these methods by incorporating tags as visible alignment indicators, leading to further performance improvements.

**Image Tagging**, also known as multi-label image recognition, is a fundamental computer vision task that involves identifying multiple tags for a given image. Traditional approaches rely on a fully connected classifier and Binary Cross-Entropy loss (BCE) for optimization. Recent studies propose transformer-based classifiers (Liu et al., 2021a; Ridnik et al., 2023) to better leverage visual features, as well as robust loss functions (Ridnik et al., 2021; Zhang et al., 2021b) to address the issues of missing samples and unbalanced positive-negative samples. Most existing multi-label datasets (Lin et al., 2014; Everingham et al., 2015) rely on manual annotations, which are labor-intensive and difficult to scale up. Our study employs text semantic parsing to efficiently obtain

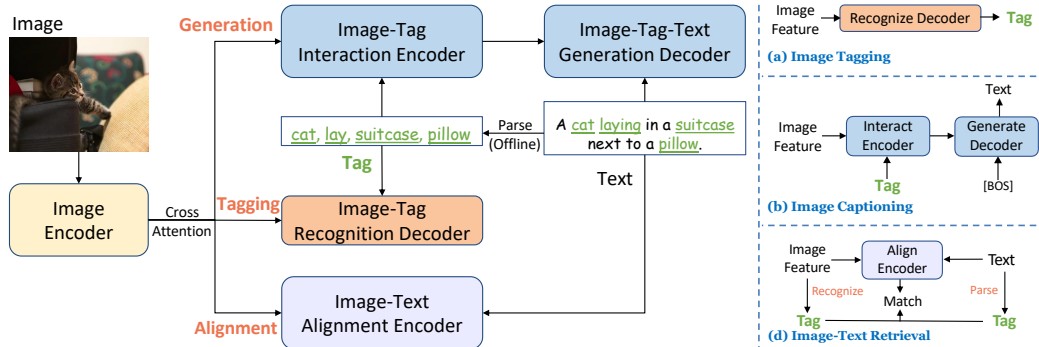

Figure 3: Illustration of Tag2Text framework. The core of Tag2Text lies in the introduction of image tagging supervised by the annotation-free image tags parsed from its paired text. **Generation**: Tag2Text learns to generate text related to the image by leveraging the automatically parsed tags, resulting in comprehensive and controllable texts with the guidance of recognized tags. **Alignment**: Tag2Text aligns the image and text, providing tags as visible alignment indicators during inference.

image tags and constructs a large-scale image tagging dataset comprising 3,429 commonly used categories, resulting in a superior tag recognition ability.

## 3 APPROACH

### 3.1 OVERVIEW FRAMEWORK

We present Tag2Text, a VLP framework that enhances the performance of vision-language models by incorporating tagging guidance. Figure 3 shows the framework of Tag2Text. With large-scale image-text pairs, the core of Tag2Text lies in the utilization of image tags from texts. Initially, the image tags are extracted through text semantic parsing, providing a large-scale of tags without expensive manual annotations. Afterward, the parsed tags can serve as ground-truth labels for image tag recognition tasks. Moreover, we design a novel scheme of image-tag-text generation, enabling the model to effectively regulate the content and quality of the generated text with the guidance of recognized tags. Furthermore, Tag2Text encompasses image-text alignment and leverages tags as visible alignment indicators.

### 3.2 MINING TAGS FROM TEXTS

**Text Semantic Parser** is adopted to parse text into image tags. The parser (Wu et al., 2019) first identifies $entities (= head + modifier)$ and $relationships$ from the input sentence based on the rules of the dependency tree, which is a grammatical structure that maps syntactic relationships within a sentence. Subsequently, we obtain the tags (including $objects$, $scenes$, $attributes$, and $actions$) of the image based on the contrast maps from $head \rightarrow object/scene$, $modifier \rightarrow attribute$, and $relationship \rightarrow action$. For instance, given the sentence "*A red alarm clock is on a wooden desk*", the parser automatically parse this as: *"head": ['alarm clock', 'desk'], "modifier": ['red', 'wooden'], "relation": ['on'].*

**Tag Category System Construction** is based on the principle that tags with higher frequency are considered more significant since they reflect common elements in the image descriptions. By employing the semantic parser, we process 4 million open-source image-text pairs and select the 5,000 most frequently occurring tags. Further filtering by human annotation results in the selection of the most commonly human-used 3,429 categories of tags (*e.g.,* synonyms such as "*person*" and "*huamn*" are merged). More statistics and details are presented in Appendix B.

### 3.3 TAG2TEXT PRE-TRAINING

With triplet image-tag-text as inputs, Tag2Text employs a multi-task pre-training approach, which consists of Tagging, Generation, and Alignment. Both generation-based and alignment-based task

utilize the guidance from image tagging to improve their performance. Concretely, the shared visual features obtained from the image encoder are interacted with various pre-training tasks through cross-attention.

**Image Tagging** aims to associate image features with the corresponding tags. We apply the image-tag recognition decoder (Liu et al., 2021a) with the robust alignment loss function for optimization. Compared to CLIP, which relies on the alignment of global image features with text via dot product interaction. Tag2Text introduces a more fine-grained alignment of visual spatial features with tags (parsed from texts) through an efficient recognition decoder. This approach is particularly effective for multi-tag recognition, since the tags often correspond to multiple image regions and reside at the token level within the texts.

**Image-Tag-Text Generation** aims to generate texts based on the image features in accordance with assigned tags. To achieve image-tag-text generation, Tag2Text employs the transformer encoder-decoder (Vaswani et al., 2017) architecture. The [BOS] token is prepended to the beginning of the text to indicate the start of a sequence. To eliminate positional bias, the image tags are rearranged prior to processing. Both tags and text are transformed into embeddings through tokenization and a word embedding matrix. The tag embeddings are integrated with image features in the image-tag interaction encoder and subsequently forwarded to the image-tag-text generation decoder for text generation. The text embeddings are utilized as ground truths to optimize the model via Language Modeling Loss (LM).

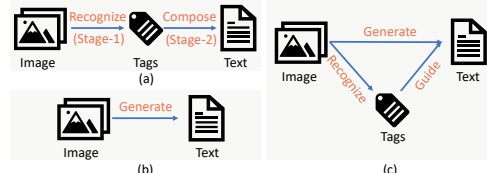

Figure 4: Image-text generation comparison. (a) Early works (Fang et al., 2015) primarily employ a multi-stage approach with separate tag recognition and text composition. The image features are not utilized during the text composition stage. (b) Recent works typically generate text directly from image features, which is challenging to control the generated text. (c) Our approach incorporates tags as a bridge to guide image features for text generation, improving content and quality control.

The distinction between image-tag-text generation and other generation approaches is illustrated in Figure 4. Our image-tag-text generation incorporates tags as a bridge to guide image features for text generation in an end-to-end manner. This approach enables the model to generate more comprehensive and controllable captions, provided that many accurate tags are given as the guidance signal.

**Image-Text Alignment** aims to determine whether a given pair of image and text is aligned. Following Li et al. (2022), Tag2Text leverages an additional image-text alignment encoder. The text is converted into embeddings via tokenization and word embedding. Then the text embeddings pass through the encoder and undergo coarse-grained Image-Text Contrastive Loss (ITC) with image features. Subsequently, the text embeddings undergo fine-grained Image-Text Matching Loss (ITM) with image features through cross-attention. The negative samples with higher ITC similarity will be selected for ITM with greater probability for hard mining.

### 3.4 TAG-GUIDED V+L TASKS

**Image Tagging**, also known as multi-label image recognition, demands the model to recognize all relevant tags of an image. Image tagging can serve as an effective indicator of the model's recognition abilities. As illustrated in Figure 3(a), Tag2Text achieves this task by directly utilizing the image-tag recognition decoder.

**Image Captioning** entails the model to generate a textual description for a given image. Figure 3(b) shows that the same components for image-tag-text generation pre-training are utilized during fine-tuning. Previous image-text generation models are challenging to control the content of the generated description. By incorporating comprehensive tags recognized by the image-tag recognition decoder, our approach effectively improves the performance over the generated text. Furthermore, users can also input alternate guidance tags to generate descriptions highlighting specific aspects of the image.

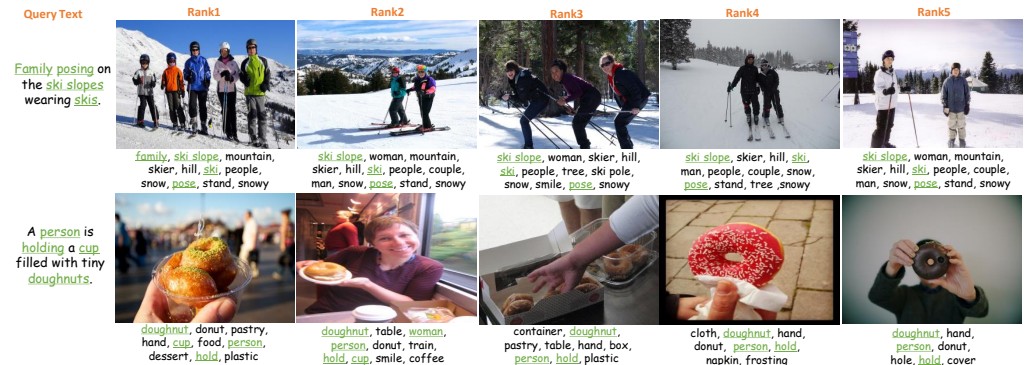

Figure 5: Example results of Tag2Text on image-text retrieval. For each query text, the top 5 instances from the retrieval set are ranked from left to right. Tag2Text provides tags as additional visible alignment indicators.

**Image-Text Retrieval** encompasses both image-to-text and text-to-image retrieval. Previous methods match image-text pairs based solely on the features of different modalities, resulting in a lack of control and interpretability. Our approach, as depicted in Figure 3(d), augments these methods by incorporating tags as visible alignment indicators ($Image \xrightarrow{Recognize} Tag, Text \xrightarrow{Parse} Tag$). By weighing the number of matching tags with feature similarity, Tag2Text boosts the retrieval results. Besides, real-world applications frequently involve users' searching for images using several keywords rather than a sentence, highlighting the advantages of our approach. Figure 5 presents some examples of visible alignment indicators that enable effective alignment between image and text.

## 4 EXPERIMENT

### 4.1 EXPERIMENTAL SETUP

Following Li et al. (2021; 2022), we pre-train our model on two widely used dataset settings, including a 4 million image dataset and a 14 million image dataset, respectively. The 4M image dataset setting includes two human-annotated datasets (COCO Lin et al. (2014) and VG Krishna et al. (2017)) and two web datasets (SBU Captions Ordonez et al. (2011) and CC-3M Sharma et al. (2018)). The 14M image dataset setting builds upon the 4M setting by adding more noisy web dataset CC-12M Changpinyo et al. (2021). We adopt two most widely used backbones pre-trained on ImageNet (Deng et al., 2009) as the image encoder: ViT$_{Base}$ (Dosovitskiy et al., 2021) and Swin$_{Base}$ (Liu et al., 2021b). Unless illustrated with subscript, the default model vision refers to Swin$_{Base}$ as the image encoder. More implementation details are provided in Appendix A.

### 4.2 EVALUATION ON IMAGE TAGGING

To assess the tagging capability of Tag2Text, we conduct evaluation on two multi-label recognition tasks: COCO and OpenImages (Kuznetsova et al., 2020). Given the significant number of rare categories with missing labels in OpenImages, we curated a subset that encompasses common categories with high quality labels. We also employed an internal high-quality annotated test set, known as OPPO, to provide a comprehensive evaluation of tagging performance. Our model is fine-tuned on the COCO training dataset using texts and tags parsed from the texts provided in COCO caption annotations, since the original COCO multi-label annotations encompass only 80 categories of tags. More details can be found in Appendix C. Addition zero-shot evaluations on NUS-WIDE (Chua et al., 2009) are provided in Appendix E.

These tagging benchmarks serve to gauge the recognition capabilities of image recognition models for prevalent categories. The comparison of Tag2Text with other SOTA recognition models (both classification models and vision-language models) is shown in Table 2. With regard to classification models, Tag2Text demonstrates superior zero-shot recognition capabilities, even comparable to full supervision manners of ML-Decoder. With regard to vision-language models, for alignment vision-language models, we calculate the similarity between an image and all tag categories with

thresholding to obtain image tags. For captioning vision-language models, we parse the caption and classify them into synonyms to obtain image tags. Remarkably, both the tagging and captioning capabilities of Tag2Text significantly exceeds other SOTA vision-language models (including CLIP, BLIP, BLIP-2) in common category recognition.

| Methods | Pre-train #Images | Evaluation Paradigm | OPPO | OpenImages | COCO |
|---|---|---|---|---|---|
| ML-Decoder (Ridnik et al., 2023) | 9M | Tagging | 82.4 | **85.8** | 72.8 |
| MKT (He et al., 2022) | 400M | Tagging | 78.2 | 77.8 | 62.9 |
| Tag2Text (Ours) | 4M | Tagging | 83.0 | 82.9 | **78.3** |
| Tag2Text (Ours) | 14M | Tagging | **85.4** | 83.4 | 78.2 |

Table 1: Performance comparison of image tagging with classification models in mAP. Blue refers to zero-shot performance; Green refers to fully supervised learning; Yellow denotes that the model has seen the corresponding training images, but not the annotations. Notably, Tag2Text's **zero-shot generalization** on OpenImages is even comparable with ML-Decoder's **full supervision**.

| Methods | Pre-train #Images | Evaluation Paradigm | OPPO | | | OpenImages | | | COCO | | |
|---|---|---|---|---|---|---|---|---|---|---|---|
| | | | F1 | Precision | Recall | F1 | Precision | Recall | F1 | Precision | Recall |
| CLIP (Radford et al., 2021) | 400M | Alignment | 63.4 | 76.6 | 54.1 | 63.0 | 77.9 | 52.9 | 48.2 | 64.0 | 38.7 |
| DiHT (Radenovic et al., 2023) | 438M | Alignment | 66.8 | 75.3 | 60.0 | 66.3 | 77.0 | 65.3 | 48.9 | 51.4 | 46.7 |
| BLIP (Li et al., 2022) | 129M | Alignment | 65.7 | 76.7 | 57.5 | 64.8 | 78.6 | 55.1 | 54.3 | 65.2 | 46.5 |
| BLIP (Li et al., 2022) | 129M | Captioning | 58.6 | 79.1 | 46.6 | 56.6 | 73.7 | 45.9 | 55.7 | 93.0 | 39.8 |
| BLIP-2 (Li et al., 2023) | 129M | Captioning | 58.2 | 72.8 | 48.5 | 58.1 | 74.2 | 47.8 | 59.1 | **95.5** | 42.8 |
| Tag2Text (Ours) | 14M | Captioning | 65.9 | **82.4** | 54.9 | 62.7 | 76.7 | 53.0 | 62.7 | 93.2 | 47.2 |
| Tag2Text (Ours) | 4M | Tagging | 75.7 | 76.6 | 74.8 | 71.8 | 79.7 | 65.3 | **72.6** | 80.5 | **66.1** |
| Tag2Text (Ours) | 14M | Tagging | **78.6** | 77.9 | **79.4** | **72.7** | **80.1** | **66.6** | 71.5 | 80.1 | 64.5 |

Table 2: Performance comparison of image tagging with vision-language models. Notably, Tag2Text showcases superior zero-shot image recognition capabilities, surpassing other vision-language models with significantly larger training dataset.

## 4.3 EVALUATION ON IMAGE CAPTIONING

In Section 4.2, we provide a novel captioning evaluation paradigm based on image tagging benchmarks, which effectively gauges the caption recognition capability for prevalent categories. In this section, we evaluate Tag2Text on two established Image Captioning benchmarks: COCO Captions (Karpathy & Fei-Fei, 2015) and NoCaps (Agrawal et al., 2019), with the latter focusing more on recognizing novel objects. The comparison of Tag2Text with other SOTA generation models can be found in Table 3. To ensure fairness, we compare the results with base version of all methods without utilizing the CIDEr optimization (Rennie et al., 2017).

The experimental results demonstrate Tag2Text outperforms other methods across all metrics on both benchmarks with similar model size and data scale. Furthermore, Tag2Text surpasses most metrics of BLIP$_{+Bootstrap}$, which employs a dataset bootstrapping approach, as well as LEMON and SIMVLM, which are pre-trained on 200 million and 1.8 billion images, respectively. Notably, due to the dual capability for both generation and alignment of Tag2Text, the performance of Tag2Text can also be further enhanced through $Bootstrap$, which we aim to accomplish in our future work.

## 4.4 EVALUATION ON IMAGE-TEXT RETRIEVAL

The Image-Text Retrieval task is evaluated on two benchmarks: COCO and Flickr30K (Plummer et al., 2015), for both image-to-text retrieval (I2T) and text-to-image retrieval (T2I). The performance comparsion with other methods are shown in Table 4. Under equivalent pre-training data and image encoder configurations, Tag2Text demonstrates comparable or superior performance compared to ALBEF, VLMO, and BLIP. Tag2Text-Swin leads to a further substantial improvement in performance. More importantly, the integration of tag alignment in Tag2Text makes it well-suited for practical search scenarios, where users search through the query of several keywords.

| Methods | Pre-train #Images | COCO Caption -Finetuning | | NoCaps Validation Zero-Shot | | | | | | | |
|---|---|---|---|---|---|---|---|---|---|---|---|
| | | | | in-domain | | near-domain | | out-domain | | overall | |
| | | B@4 | C | C | S | C | S | C | S | C | S |
| *Pre-trained with 4M images (COCO, VG, SBU, CC-3M):* | | | | | | | | | | | |
| DistillVLM (Fang et al., 2021) | 4M | 35.6 | 120.8 | - | - | - | - | - | - | - | - |
| UFO (Wang et al., 2021a) | 4M | 36.0 | 122.8 | 94.5 | 13.4 | 82.7 | 12.8 | 64.9 | 11.0 | 80.7 | 12.5 |
| OSCAR (Li et al., 2020b) | 4M | 36.5 | 123.7 | 80.0 | 12.1 | 80.4 | 12.2 | 75.3 | 10.6 | 79.3 | 11.9 |
| VIVO (Hu et al., 2021) | 4M | - | - | 90.4 | 13.0 | 84.9 | 12.5 | 83.0 | 10.7 | 85.3 | 12.2 |
| BLIP* (Li et al., 2022) | 4M | 37.0 | 123.6 | 91.3 | 13.5 | 86.4 | 13.0 | 84.2 | 11.8 | 86.6 | 12.8 |
| ViTCap (Fang et al., 2022) | 4M | 36.3 | 125.2 | 98.7 | 13.3 | 92.3 | 13.3 | 95.4 | 12.7 | 93.8 | 13.0 |
| Tag2Text-ViT (Ours) | 4M | 37.3 | 124.6 | 95.1 | 13.5 | 89.0 | 13.0 | 86.9 | 12.1 | 89.5 | 12.9 |
| Tag2Text-Swin (Ours) | 4M | **38.4** | **128.7** | **101.0** | **13.9** | **96.0** | **13.6** | **96.7** | **12.9** | **96.9** | **13.5** |
| *Pre-trained with more images:* | | | | | | | | | | | |
| Enc-Dec (Changpinyo et al., 2021) | 15M | - | 110.9 | 92.6 | 12.5 | 88.3 | 12.1 | 94.5 | 11.9 | 90.2 | 12.1 |
| MiniVLM (Wang et al., 2020) | 11M | 35.6 | 119.8 | - | - | - | - | - | - | - | - |
| VinVL (Zhang et al., 2021a) | 6M | 38.2 | 129.3 | 103.7 | 13.7 | 95.6 | 13.4 | 83.8 | 11.9 | 94.3 | 13.1 |
| LEMON (Hu et al., 2022) | 12M | - | - | 104.5 | 14.6 | 100.7 | 14.0 | 96.7 | 12.4 | 100.4 | 13.8 |
| BLIP (Li et al., 2022) | 14M | 38.0 | 127.8 | - | - | - | - | - | - | 102.2 | 13.9 |
| Tag2Text-ViT (Ours) | 14M | 38.4 | 128.9 | 104.8 | **14.6** | 100.4 | 13.9 | 102.5 | 13.3 | 101.5 | 14.0 |
| Tag2Text-Swin (Ours) | 14M | **39.1** | **131.8** | **106.7** | 14.5 | **105.7** | **14.4** | **110.7** | **14.3** | **106.9** | **14.4** |
| *Addition Data Augmentation or Higher-Scale Datasets:* | | | | | | | | | | | |
| BLIP+$_{Bootstrap}$ (Li et al., 2022) | 14M | 38.6 | 129.7 | 111.3 | 15.1 | 104.5 | 14.4 | 102.4 | 13.7 | 105.1 | 14.4 |
| SIMVLM (Wang et al., 2021b) | 1.8B | 39.0 | 134.8 | - | - | - | - | - | - | 94.8 | 13.1 |
| LEMON (Hu et al., 2022) | 200M | - | - | 107.7 | 14.7 | 106.2 | 14.3 | 107.9 | 13.1 | 106.8 | 14.1 |

Table 3: Performance comparison of image captioning on the COCO and NoCaps Caption benchmarks. BLIP* refers to the result of our reproduction. $_{+Bootstrap}$ indicates the two stage dataset bootsrapping approach using generation and alignment tasks.

| Methods | Pre-train #Images | COCO(5K test set) | | | | | | Flickr30K(1K test set) | | | | | |
|---|---|---|---|---|---|---|---|---|---|---|---|---|---|
| | | I2T | | | T2I | | | I2T | | | T2I | | |
| | | R@1 | R@5 | R@10 | R@1 | R@5 | R@10 | R@1 | R@5 | R@10 | R@1 | R@5 | R@10 |
| *Pre-trained with 4M images (COCO, VG, SBU, CC-3M):* | | | | | | | | | | | | | |
| UNITER (Chen et al., 2020) | 4M | 65.7 | 88.6 | 93.8 | 52.9 | 79.9 | 88.0 | 87.3 | 98.0 | 99.2 | 75.6 | 94.1 | 96.8 |
| VILLA (Gan et al., 2020) | 4M | - | - | - | - | - | - | 87.9 | 97.5 | 98.8 | 76.3 | 94.2 | 96.8 |
| OSCAR (Li et al., 2020b) | 4M | 70.0 | 91.1 | 95.5 | 54.0 | 80.8 | 88.5 | - | - | - | - | - | - |
| METER-Swin(Dou et al., 2022) | 4M | 73.0 | 92.0 | 96.3 | 54.9 | 81.4 | 89.3 | 94.3 | 99.6 | 99.9 | 82.2 | 96.3 | 98.4 |
| ALBEF(Li et al., 2021) | 4M | 73.1 | 91.4 | 96.0 | 56.8 | 81.5 | 89.2 | 94.3 | 99.4 | 99.8 | 82.8 | 96.7 | 98.4 |
| VLMO (Bao et al., 2021) | 4M | 74.8 | 93.1 | 96.9 | 57.2 | 82.6 | 89.8 | 92.3 | 99.4 | 99.9 | 79.3 | 95.7 | 97.8 |
| BLIP* (Li et al., 2022) | 4M | 75.0 | 92.7 | 96.2 | 56.9 | 81.9 | 88.9 | **95.0** | 99.6 | 99.9 | 81.9 | 96.0 | 98.0 |
| OmniVL (Wang et al., 2022b) | 4M+Videos | 76.8 | 93.6 | **97.3** | 58.5 | 82.6 | 89.5 | 94.9 | 99.4 | 99.9 | 83.4 | **97.0** | **98.6** |
| Tag2Text-Vit (Ours) | 4M | 74.9 | 92.5 | 96.2 | 56.6 | 81.5 | 88.8 | 94.3 | 98.9 | 99.6 | 80.5 | 95.5 | 97.6 |
| Tag2Text-Swin (Ours) | 4M | **77.5** | **94.1** | 97.2 | **60.0** | **83.3** | **89.9** | 94.8 | **99.6** | **100.0** | **84.2** | 96.7 | 98.5 |
| *Pre-trained with more images:* | | | | | | | | | | | | | |
| FLAVA (Singh et al., 2022) | 70M | 61.5 | 82.1 | 89.6 | 50.1 | 74.4 | 83.2 | 85.4 | 95.7 | 98.3 | 73.2 | 92.7 | 95.5 |
| UNIMO (Li et al., 2020a) | 5.7M | - | - | - | - | - | - | 89.4 | 98.9 | 99.8 | 78.0 | 94.2 | 97.1 |
| METER-CLIP-ViT (Dou et al., 2022) | 404M | 76.2 | 93.2 | 96.8 | 57.1 | 82.7 | 90.1 | 94.3 | 99.6 | 99.9 | 82.2 | 96.3 | 98.4 |
| ALIGN (Jia et al., 2021) | 1.8B | 77.0 | 93.5 | 96.9 | 59.9 | 83.3 | 89.8 | 95.3 | 99.8 | **100.0** | 84.9 | 97.4 | 98.6 |
| ALBEF(Li et al., 2021) | 14M | 77.6 | 94.3 | 97.2 | 60.7 | 84.3 | 90.5 | 95.9 | 99.8 | 100.0 | 85.6 | 97.5 | 98.9 |
| BLIP (Li et al., 2022) | 14M | 78.4 | - | - | - | - | - | - | - | - | - | - | - |
| Tag2Text-Vit (Ours) | 14M | 77.8 | 93.9 | 97.0 | 59.3 | 83.1 | 89.7 | 94.9 | 99.7 | **100.0** | 83.3 | 96.4 | 98.1 |
| Tag2Text-Swin (Ours) | 14M | **79.0** | **94.6** | **97.2** | **61.3** | **84.3** | 90.4 | 95.7 | **99.9** | **100.0** | 85.4 | 96.8 | 98.5 |
| BLIP+$_{Bootstrap}$ (Li et al., 2022) | 14M | 80.6 | 95.2 | 97.6 | 63.1 | 85.3 | 91.1 | 96.6 | 99.8 | 100.0 | 87.2 | 97.5 | 98.8 |

Table 4: Performance comparison of image-text retrieval on the COCO and Flickr30K benchmarks. BLIP* refers to the result of our reproduction. $_{+Bootstrap}$ indicates the two stage dataset bootsrapping approach using generation and alignment tasks.

## 4.5 ANALYSIS OF TAGGING GUIDANCE

In this section, we present a detailed analysis to investigate the effectiveness of tagging guidance.

**Evaluation of Tagging Guidance.** In Table 5, we verify the superiority of incorporating tagging guidance on a wide range of downstream benchmarks, including four generation benchmarks, two retrieval benchmarks, and two recognition benchmarks.

| Methods | Pre-train #Images | Image Captioning | | | | | | Image-Text Retrieval | | | | Image Tagging | |
|---|---|---|---|---|---|---|---|---|---|---|---|---|---|
| | | OPPO-ZS | | OpenImages-ZS | | COCO-FT | NoCaps-ZS | COCO-FT | | Flickr-ZS | | COCO-FT | OpenImages-ZS |
| | | Precision | Recall | Precision | Recall | CIDEr | CIDEr | TR@1 | IR@1 | TR@1 | IR@1 | mAP | mAP |
| w/o Tag Guidance | 4M | 78.5 | 45.3 | 72.0 | 44.6 | 123.3 | 88.8 | 74.4 | 56.1 | 90.0 | 74.5 | ✗ | ✗ |
| w Tag Guidance | 4M | $81.4_{+2.9}$ | $52.0_{+6.7}$ | $75.1_{+3.1}$ | $52.0_{+7.4}$ | $124.6_{+1.3}$ | $89.5_{+0.7}$ | $74.9_{+0.5}$ | $56.6_{+0.5}$ | $90.4_{+0.4}$ | $75.2_{+0.7}$ | 78.3 | 82.9 |
| w Tag Guidance | 14M | 82.4 | 54.9 | 76.7 | 53.0 | 128.9 | 101.5 | 77.8 | 59.3 | 92.7 | 78.7 | 78.2 | 83.4 |

Table 5: Evaluation of tagging guidance on eight downstream benchmarks with finetuning (FT) and zero-shot (ZS) settings. ✗ refers to the method which cannot be directly transferred to the corresponding benchmark.

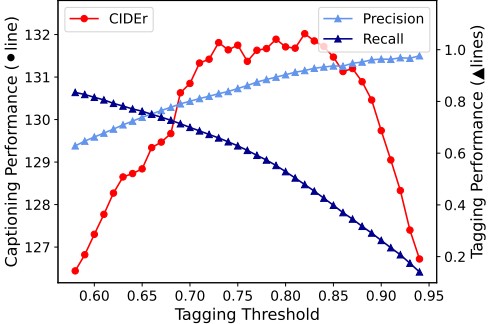

Figure 6: The strong correlation between captioning performance and tagging guidance performance of tag2text demonstrates that tagging guidance exerts significant control over image captioning. ● lines with the left axis: image captioning performance. ▲ lines with the right axis: tagging guidance performance.

Figure 7: The comparison of recognized tags between Tag2Text and other SOTA models for multi-label recognition (ML-Decoder Ridnik et al. (2023)) and object detection (Detic Zhou et al. (2022)). Tag2Text offers more comprehensive and commonly used tags including objects, scenes, attributes, and actions.

**Controllability Analysis.** We provide the analysis of the controllability of tagging guidance for image captioning. We manipulate the threshold of the tagging head to obtain tagging guidance of varying quality. As depicted in Figure 6, the captioning performance (evaluation on COCO) declines when the precision or recall of tagging (evaluation on OpenImages) is low. These results effectively establish that tagging guidance exerts significant control over image captioning.

**Better Bridge between Image and Text.** In order to highlight the superiority of Tag2Text in tag recognition, we compare the recognized tags with other SOTA open-source models on multi-label recognition and object detection. For multi-label recognition, we employ the ML-Decoder (Ridnik et al., 2023) model based on OpenImages (Kuznetsova et al., 2020) of 9,600 categories. For object detection, we employ the Detic (Zhou et al., 2022) model based on LVIS (Gupta et al., 2019) of 1,203 categories. The comparison results are illustrated in Figure 7, ML-Decoder recognizes many tags which are not frequently used and lacks many obvious common tags. On the other hand, Detic is limited to only recognizing object categories. In contrast, Tag2Text provides a more comprehensive and widely used set of tags, including objects, scenes, attributes, and actions.

**Ablation Study.** Despite the presence of noise for tags parsed from the texts, our model design enables Tag2Text to leverage tags with noise and achieve exceptional image tagging performance. As demonstrated in Table 6, the integration of vision-language pre-training tasks into the model also improves the tag recognition ability. Furthermore, Table 6 highlights two-stage "pre-training + finetuning" paradigm in the context of multi-label recognition. The model, trained solely on the limited COCO dataset, fails to generalize well on the OpenImages dataset, attaining an mAP score of 57.5.

| Methods | Pre-train #Images | Finetune on COCO | COCO mAP | OpenImages mAP |
|---|---|---|---|---|
| *Training with full annotations of image tags:* | | | | |
| ML-Decoder | - | - | 72.8 | 85.8 |
| *Training with image-text pairs:* | | | | |
| Tagging | 4M | ✗ | 74.5 | 74.7 |
| Tag2Text | 4M | ✗ | 74.7 | 76.4 |
| Tag2Text | ✗ | ✓ | 75.6 | 57.5 |
| Tag2Text | 4M | ✓ | 78.3 | 82.9 |
| Tag2Text | 14M | ✓ | 78.2 | 83.4 |

Table 6: Ablation study on image tagging. The representation of background color is consistent with Table 2.

However, when pre-trained on a large dataset, our model exhibits remarkable performance, even in the absence of any exposure to the training images from the OpenImages dataset, achieving an mAP score of 83.4, which is comparable to the fully supervised performance of 85.8 mAP.

## 5 CONCLUSION

This paper has presented Tag2Text, a vision-language pre-training framework, which introduces image tagging into vision-language models. Tag2Text achieves superior image tag recognition ability by exploiting fine-grained text information. Moreover, Tag2Text leverages tagging guidance and effectively enhances the performance and controllability of vision-language models. On a wide range of vision-language tasks, Tag2Text demonstrates the value of tag as a bridge between image and text to infuse structure and knowledge information into vision-language models.

ACKNOWLEDGMENTS

This work was supported by the National Natural Science Foundation of China (No. 62172101), the Science and Technology Commission of Shanghai Municipality (No.22DZ1100101, No. 21511100500), and the OPPO Research Foundation.

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

# A  PRE-TRAINING DETAILS

## A.1  IMPLEMENTATION DETAILS.

The encoder-decoder used for text generation and encoder for image-text alignment are 12-layer transformers (Vaswani et al., 2017) initialized from $\text{BERT}_{Base}$ (Devlin et al., 2018). The generation decoder and alignment encoder share parameters with the cross-attention layers. The tag recognition decoder is a 2-layer transformer initialized from $\text{BERT}_{Base}$ and shares parameters with the lowest 2-layer of the interaction encoder. The models are pre-trained for 20 epochs with the batch size of 960 on 8 NVIDIA A100 GPUs. The optimizer is the AdamW (Loshchilov & Hutter, 2017) with a weight decay of 0.05. The learning rate is warmed-up to $1e^{-4}$ in the first 3,000 iterations, and then follows linear decay with a rate of 0.9. The input images are resized to $224 \times 224$ uniformly during the pre-training stage. Due to the presence of missing labels in the parsed tags and an imbalanced distribution of positive and negative samples, we employ Asymmetric Loss (ASL) (Ridnik et al., 2021) for image tagging optimization.

## A.2  PRE-TRAINING OBJECTIVES.

**Image Tagging.** Image tagging is generally decomposed into multiple binary classification with binary cross-entropy loss (BCE) to optimize:

$$
\begin{aligned}
\mathcal{L}_{\text{Tagging}} &= -\mathbb{E}_{\mathbf{y} \sim D}\left[BCE(\mathbf{y}, P(\mathbf{y}))\right] \\
&= -\mathbb{E}_{\mathbf{y} \sim D}\left[\sum_{i=1}^{C} \mathbf{y}_i \log P\left(\mathbf{y}_i\right) + (1 - \mathbf{y}_i) \log(1 - P\left(\mathbf{y}_i\right))\right]
\end{aligned}
\tag{1}
$$

where $y_i$ represents the label for the $i$-th category and $C$ denotes the total number of categories. We employ Asymmetric Loss (ASL) (Ridnik et al., 2021) for optimization instead of BCE.

$$
\begin{aligned}
\mathcal{L}_{\text{Tagging}} &= -\mathbb{E}_{\mathbf{y} \sim D}\left[ASL(\mathbf{y}, P(\mathbf{y}))\right] \\
&= -\mathbb{E}_{\mathbf{y} \sim D}\Big[\sum_{i=1}^{C} \mathbf{y}_i (1 - P\left(\mathbf{y}_i\right))^{\gamma_+} \log P\left(\mathbf{y}_i\right) \\
&\quad + (1 - \mathbf{y}_i) P\left(\mathbf{y}_i\right)^{\gamma_-} \log(1 - P\left(\mathbf{y}_i\right))\Big]
\end{aligned}
\tag{2}
$$

**Image-Tag-Text Generation.** The pre-training objective for image-tag-text generation is Language Modeling Loss (LM) (Brown et al., 2020) to maximize the likelihood of the text in an autoregressive manner:

$$
\begin{aligned}
\mathcal{L}_{\text{LM}} &= -\mathbb{E}_{\mathbf{x} \sim D}\left[CE(\mathbf{x}, P(\mathbf{x}))\right] \\
&= -\mathbb{E}_{\mathbf{x} \sim D}\left[\sum_{i=1}^{N} \log P\left(\mathbf{x}_i \mid \mathbf{x}_{<i}\right)\right]
\end{aligned}
\tag{3}
$$

where $x_i$ represents the $i$-th token in the text and $N$ denotes the total number of text tokens. Compared with the bidirectional Masked Language Modeling (MLM) (Devlin et al., 2018), the unidirectional LM Loss is also gradually widely used in recent VLP studies (Wang et al., 2021b; 2022a; Chen et al., 2022; Li et al., 2022), as it enables seamless transfer of the model to text generation tasks.

**Image-Text Alignment.** The pre-training objectives for alignment utilizes Image-Text Contrastive Loss (ITC) based on multi-modal feature cos similarity (Radford et al., 2021; Li et al., 2021; Bao et al., 2021; Li et al., 2022) and Image-Text Matching Loss (ITM) based on multi-modal feature fusion (Li et al., 2021; Bao et al., 2021; Li et al., 2022).

$$\mathcal{L}_{\text{ITC}} = -\mathbb{E}_{\mathbf{I},\mathbf{T}\sim D}\left[CE(\mathbf{y}(\mathbf{I}), P(\mathbf{I})) + CE(\mathbf{y}(\mathbf{T}), P(\mathbf{T}))\right] \tag{4}$$

$$\mathcal{L}_{\text{ITM}} = -\mathbb{E}_{\mathbf{I},\mathbf{T}\sim D}\left[BCE(\mathbf{y}, P(\mathbf{I}, \mathbf{T}))\right] \tag{5}$$

# B  TAG CATEGORY DETAILS

**Pre-training Dataset.** The pre-training dataset statistics, including the number of texts and tags parsed from texts, are presented in Table 7.

|            | COCO | VG   | SBU   | CC-3M | CC-12M |
|------------|------|------|-------|-------|--------|
| #images    | 113K | 100K | 849K  | 2.81M | 10.26M |
| #texts     | 567K | 769K | 849K  | 2.81M | 10.26M |
| #avg texts | 5.02 | 7.69 | 1.00  | 1.00  | 1.00   |
| #tags      | 791K | 607K | 1.56M | 5.93M | 23.16M |
| #avg tags  | 7.00 | 6.07 | 1.84  | 2.11  | 2.26   |

Table 7: The statistics of the pre-training datasets.

**Tag Category System.** We obtain annotation-free image tags parsed from its paired text. We construct our tag category system based on the principle that tags with a higher frequency are more significant, as they represent common elements in image descriptions. To this end, we process 4 million open-source image-text pairs (COCO, VG, SBU, CC-3M), and select the 5,000 most frequently occurring tags. Further filtering by human annotation results in the selection of the most commonly human-used 3,429 categories of tags (including objects, scenes, attributes, actions). The tag category statistics are presented in Table 8. An illustration of the tag categories is provided in Figure 8, where the size of each word is proportional to the frequency of the category in the open-source image-text pairs.

|            | Object/Scene | Attribution | Action | Total |
|------------|--------------|-------------|--------|-------|
| Categories | 3,012        | 177         | 240    | 3429  |

Table 8: The statistics of tag categories recognized by Tag2Text.

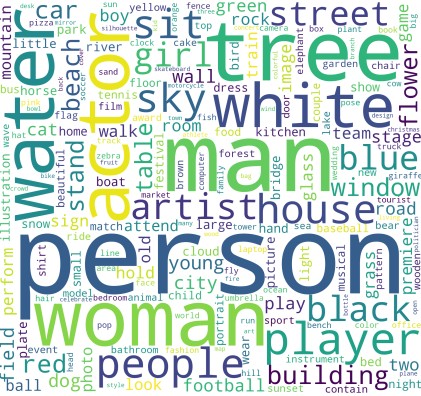

Figure 8: Illustration of the most-frequent categories in our tag category system. The word size is proportional to the category frequency in the pre-training image-text pair dataset.

**Comparison with Public Datasets.** This section provides the statistics of the overlap between our tag categories and other widely used public datasets. Table 9 shows the statistics for object/scene categories with other datasets including OpenImages (Kuznetsova et al., 2020), COCO (Lin et al., 2014), ImageNet (Deng et al., 2009), CIFAR100 (Krizhevsky et al., 2009). Table 10 presents

the statistics for action categories with other datasets including HICO (Chao et al., 2015), V-COCO (Gupta & Malik, 2015), HOI-W (Liao et al., 2020). To the best of our knowledge, we do not find appropriate public datasets for the recognition of attribute tag categories.

| Categories | OpenImages | COCO | ImageNet | CIFAR100 |
|---|---|---|---|---|
| Original | 19982 | 91 | 1000 | 120 |
| Overlapping | 1988 | 73 | 358 | 94 |

Table 9: The statistics of object/scene categories overlapping with other public datasets.

| Categories | HICO | V-COCO | HOI-W |
|---|---|---|---|
| Original | 117 | 23 | 9 |
| Overlapping | 90 | 26 | 9 |

Table 10: The statistics of action categories overlapping with other public datasets.

**Impact on Vocabulary Set Size.** In Table 11, we expand the vocabulary set from 3,429 to 4,585 categories and compare the performances. Notably, the tagging performance decrease with the larger vocabulary set. We attribute to two possible reasons: *1)* The increased complexity in training with more categories. *2)* The additional categories leading to more noise, as they lack sufficient training data, thereby impacting the model's efficiency.

| Vocabulary Set Size | OPPO | OpenImages |
|---|---|---|
| 3,429 | 81.7 | 84.1 |
| 4,585 | 80.3 | 83.1 |

Table 11: The statistics of action categories overlapping with other public datasets.

## C IMAGE TAGGING DETAILS

**Tuning Dataset.** We employ tags parsed from COCO Caption (Lin et al., 2014) for image tagging finetuning. Each image in the COCO Caption dataset is accompanied by five descriptive sentences, offering a comprehensive description of the image. As a result, the tags parsed from these captions are considered to be a close approximation of a complete set of tag labels.

**Test Benchmarks.** We respectively take the overlapping categories of Tag2Text with the tagging benchmarks for evaluation. The statistics of the image tagging benchmarks set are shown in Table 12.

**Tagging Head Comparison.** Table 13 investigates the impact of various tagging recognition heads on the performance of Tag2Text. The results show that transitioning from full connection to tag recognition decoder (Liu et al., 2021a) results in improved performance in image tagging recognition, followed by improvements in caption generation. This indicates that the enhancement of tagging recognition leads to improved text generation. To mitigate the increase in model parameters, we propose sharing the parameters of image-tag recognition decoder and image-tag interaction encoder, reducing the parameters and further boosting the performance.

**Control of Tagging Guidance.** During the image tagging inference process, the tagging head outputs logits (ranging from 0 to 1) for each category. These logits are compared to a set threshold to determine the output tags. When the logits exceed this threshold, the corresponding tag category is outputted. Therefore, the tagging guidance can be controlled by adjusting the threshold. For instance, a lower threshold yields more image tags, resulting in higher recall. On the contrary, a higher threshold increases precision.

| Benchmark | #Category | #Images |
|---|---|---|
| OPPO | 200 | 44,606 |
| OpenImages | 214 | 57,224 |
| COCO | 80 | 5,000 |
| NUS-WIDE | 81 | 50,720 |

Table 12: The statistics of image tagging test benchmarks.

| Recognition Head | #Parameters | Caption-FT (COCO) | Caption-ZS (NoCaps) | Tagging-ZS (OpenImages) |
|---|---|---|---|---|
| Full Connection | 392M | 120.6 | 86.6 | 79.8 |
| Recognition Decoder | 409M | 121.2 | 87.0 | 81.2 |
| Recognition Decoder (Layer Shared) | 394M | 121.6 | 87.2 | 81.9 |

Table 13: Tagging recognition head comparison.

## D    IMAGE CAPTIONING DETAILS

**Finetuning Strategies Comparison.** This section discusses two strategies employed by Tag2Text for image captioning finetuning based on input guidance tags, as illustrated in Figure 9 ① and ②. The first strategy, similar to image-tag-text generation in the pre-training stage, involves input guidance tags parsed from the paired text. This enables the model to utilize all available tags to create a comprehensive text description. However, Tag2Text usually recognizes tags with similar meanings (*e.g.,* "man", "person"), which may result in redundant sentences (*e.g.,* "a man ..., while a person ...") with low evaluation metrics.

The second strategy involves using the same process as the inference stage, where the input guidance tags for fine-tuning are recognized by the model. This approach allows the model to select guidance tags for generating more precise text generation. In this paper, we utilize a mixed training approach that combines both strategies.

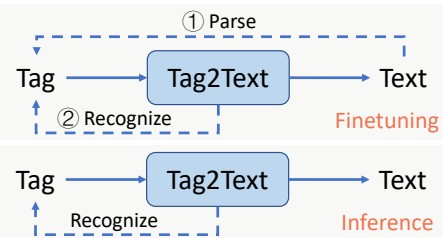

Figure 9: Illustration of two finetuning strategies for Image Captioning. By controlling the ratio of the two strategies, Tag2Text can generate more comprehensive (①) or more accurate (②) descriptions.

**More Example Results.** In Figure 12, we show more examples of Tag2Text using tag guidance to generate comprehensive descriptions.

## E    ADDITION ZERO-SHOT EVALUATIONS

In this section, we conduct addition zero-shot evaluations on NUS-WIDE Chua et al. (2009), a well-established tagging benchmark including 81 categories. All images in NUS-WIDE are out-of-distribution data, since Tag2Text did not utilize any NUS-WDIE training images during its training process. The results are presented in the Table 14. Notably, Tag2Text also demonstrates superior zero-shot performance, exceeding both CLIP and BLIP, while utilizing much less training data.

| Methods | Pre-train #Images | Evaluation Paradigm | F1 | Precision | Recall |
|---|---|---|---|---|---|
| CLIP | 400M | Alignment | 46.0 | 54.0 | 40.1 |
| BLIP | 129M | Alignment | 45.0 | 54.0 | 38.6 |
| Tag2Text | 14M | Tagging | **46.4** | **54.7** | **40.3** |

Table 14: Zero-shot Performance Comparison on NUS-WIDE.

## F   EVALUATION ON VISUAL QUESTION ANSWERING

**Visual Question Answering** aims to predict an answer to a question based on an image. Previous approaches typically treat VQA as a multi-class classification problem with a fixed set of answer choices. In contrast, Tag2Text employs an encoder-decoder architecture for generation, which is suited for generating free-form answers. As shown in Figure 3(c), the question, joint with tags, interacts with image features in the encoder, and then forwards to the decoder to generate a free-form answer.

We conduct experiments on the VQA v2 (Goyal et al., 2017) benchmark for Visual Question Answering. Table 11 shows that Tag2Text outperforms or achieves competitive results with other approaches. On the one hand, Tag2Text-ViT also ought not to be inferior to ALBEF, as the structure of Tag2Text degenerates into that of ALBEF without tagging guidance. We attribute the slightly inferior performance to our insufficient resources available for conducting hyper-parameter search.

On the other hand, we observe that the VQA v2 dataset is characterized primarily by straightforward questions and answers (*e.g.,* "Is there a big tree behind the clock? Yes."), which is challenging to directly augment through identified tags. We anticipate that a more strategic utilization of fine-grained positioning information derived from tagging guidance, combined with more complex benchmarks, can effectively highlight the superiority of tagging guidance for VQA tasks. We leave these explorations for future research.

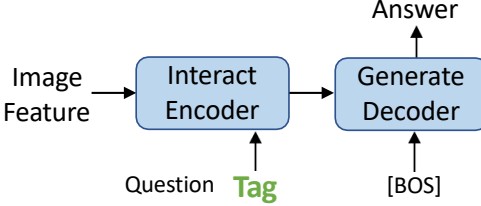

Figure 10: Illustration of Tag2Text on VQA finetuning.

| Method | Pre-train #Images | test-dev | test-std |
|---|---|---|---|
| ViLT (Kim et al., 2021) | 4M | 71.26 | - |
| FLAVA (Singh et al., 2022) | 68M | 72.80 | - |
| UNITER (Chen et al., 2020) | 4M | 72.70 | 72.91 |
| OSCAR (Li et al., 2020b) | 4M | 73.16 | 73.44 |
| VILLA (Gan et al., 2020) | 4M | 73.59 | 73.67 |
| UNIMO (Li et al., 2020a) | 5.7M | 75.06 | 75.27 |
| ALBEF (Li et al., 2021) | 14M | **75.84** | **76.04** |
| Tag2Text-ViT | 14M | 75.32 | 75.41 |
| Tag2Text-Swin | 14M | **75.84** | 75.82 |

Figure 11: Performance comparison on the VQA v2 benchmark.

## G   LIMITATIONS

**Hallucinatory Captions.** Tag2Text benefits from its powerful tagging capabilities. As depicted in Figure 7, there is a strong correlation between captioning performance and tagging guidance performance. In practical applications, we observe that incorrect user-provided tags may lead to hallucinatory captions.

**Small Objects.** In addition, evaluating a tagging model capable on 3,429 categories is also challenging. Our quantitative comparison and visual validations reveal that Tag2Text efficiently recognizes common objects and scenes, yet struggles in small objects (e.g., spoon or baseball). Our empirical experiments indicates that increasing the resolution during fine-tuning significantly improves performance on these small objects.

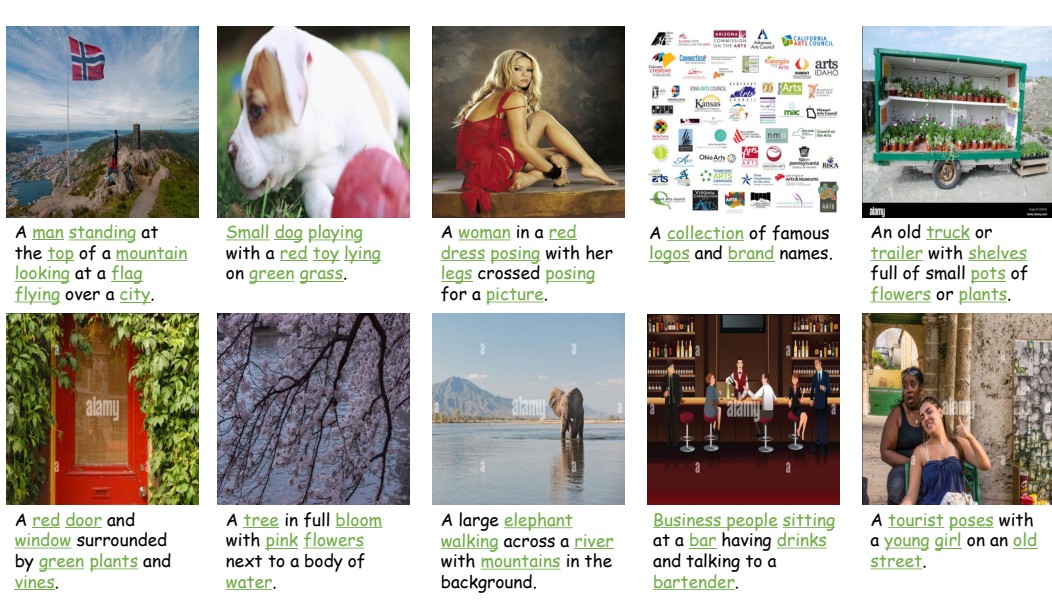

Figure 12: More image captioning results. Tag2Text integrates recognized image tags into text generation as guiding elements (highlighted in green underline), resulting in the generation with comprehensive text descriptions.

