# OpenReview forum: "Tag2Text: Guiding Vision-Language Model via Image Tagging"
_ICLR.cc/2024/Conference — ICLR 2024 poster_

### Official Review · Reviewer_Nwx3 · 2023-10-27

**Soundness:** 4 excellent
**Presentation:** 4 excellent
**Contribution:** 4 excellent
**Rating:** 8
**Confidence:** 5

**Summary:**

The paper proposes the use of tags as part of an otherwise standard V&L model. Tags are incorporated by parsing the captions to produce tags within a vocabulary, and then simply predicting the tags in a standard fashion. Secondly, tags are used, in combination with image features, to predict the caption text in an autoregressive manner. Good results are shown for tagging and for captioning, while some level of control over the generated text by using input tags is also obtained.

**Strengths:**

I really like the paper. It is clearly motivated, well written, technically well executed, has clear novelty, and has good experimental results. But I want to highlight one thing that is surprisingly hard to find: it seems useful. The reasons: 1) Current V&L models are trained with long captions, which means small tag-like queries are not well supported. There are indeed some object detection methods a-la CLIP (owl-vit and co) but they offer different functionalities and have different requirements. So this is a good addition to the V&L toolbox. 2) it offers a way of controlling caption generation through the use of input tags.


Minor suggestions (up to the authors and no reply needed):
Fig. 1 might get a bit confusing as both the "prior work strategy" and the current strategy are included in the same flow graph.
Table 2 shows the last 3 methods seem apart from the rest but it's unclear why they are separated.

**Weaknesses:**

The method is based on some relatively standard techniques that are however well executed and put together. While not much of a minus, but maybe a reason for an accept vs strong accept.

**Questions:**

1) The tag system works based on pre-existing vocabulary. What happens with out-of-vocabulary terms?
2) Have you tested retrieval based on short queries? E.g. retrieval using a term within the vocabulary vs some short retrieval query that is not part of the vocabulary (e.g. adjective + noun) vs retrieval with long captions typical of flickr/coco

---

> ### Author Response · Authors · 2023-11-13
> **Response to Reviewer Nwx3**
>
> Thanks for your encouraging words and constructive comments!
>
> We sincerely appreciate your time in reviewing the paper, and our point-to-point responses to your comments are given below.
>
>
> ### Additional Highlight Contributions ###
>
> > *"I really like the paper. It is clearly motivated, well written, technically well executed, has clear novelty, and has good experimental results. But I want to highlight one thing that is surprisingly hard to find: it seems useful. The reasons: 1) Current V&L models are trained with long captions, which means small tag-like queries are not well supported. There are indeed some object detection methods a-la CLIP (owl-vit and co) but they offer different functionalities and have different requirements. So this is a good addition to the V&L toolbox. 2) it offers a way of controlling caption generation through the use of input tags. "*
>
>
> In response to your valuable insights:
>
> * (1) We share the same viewpoint that current V&L models fall short in handling small tag-like queries. By simultaneously supporting both tagging and V&L tasks, Tag2Text significantly extends its applicability in the field.
>
>
> * (2) The flexibility of Tag2Text to integrate tagging guidance, either comprehensive recognized image tags or any user-specified tags, indeed offers an enhanced method for controlling caption generation.
>
>
> We have refined the caption of Figure 2, and will carefully incorporate these insights into our revised manuscript to further highlight its contributions to the field. Thanks very much!
>
>
> ### Clarifications on Table 2 ###
>
> > *"Table 2 shows the last 3 methods seem apart from the rest but it's unclear why they are separated."*
>
> Thanks for the constructive suggestions.
>
> The last three methods in Table 2 are distinct due to their use of pseudo-label data augmentation or higher-scale pre-training data. We have refined the presentation of Table 3 in the revision.
>
>
> ### Handling Out-of-Vocabulary Terms ###
>
> > *"The tag system works based on pre-existing vocabulary. What happens with out-of-vocabulary terms? "*
>
> In response to your query about out-of-vocabulary terms:
>
>
> * **Captioning Impact.** Our findings indicate that the out-of-vocabulary tags do not affect the caption resulting. We analyze the reason is that only tags within the vocabulary are utilized for the Image-Tag-Text generation training task.
>
> * **Out-of-Vocabulary Recognition.** For the image tagging branch, Tag2Text remains in the pre-existing vocabulary of 3,429 categories, unable to recognize out-of-vocabulary categories. Nevertheless, for out-of-vocabulary recognition, the image-text alignment branch can be utilized. We anticipate its performance to be similar with CLIP and BLIP under the same training dataset.
>
>
> * **Future Enhancements.** We are exploring enhancements to advance Tag2Text into an stronger open-set tagging model in the future works. One potential research route involves employing an off-the-shelf text encoder to encode pre-existing tag embeddings for training, enabling it to generalize open-vocabulary tag embeddings based on embedding similarity during inference. Current experiments in this direction have yielded effective results.
>
>
>
> ### Retrieval with Short Queries ###
>
> > *" Have you tested retrieval based on short queries? E.g. retrieval using a term within the vocabulary vs some short retrieval query that is not part of the vocabulary (e.g. adjective + noun) vs retrieval with long captions typical of flickr/coco "*
>
>
> Thank you for raising the aspect of retrieval with short queries, which is indeed a practical and valuable scenario.
>
>
> * **Lack of Benchmarks.** However, there is currently a lack of established benchmarks for short query retrieval. Our experiment attempts of only using short query tags indicated inferior performance compared to longer captions on the Flickr benchmark.
>
>
> * **Real-World Evaluation.** We have conducted internal evaluations with practical examples. Our findings demonstrate that when handling short queries including in-vocabulary terms, Tag2Text significantly outperforms CLIP and BLIP. However, for short queries involving out-of-vocabulary terms, Tag2Text did not show a marked improvement.
>
>
> * **Future Enhancements.** In light of these findings, we are committed to enhancing Tag2Text's efficacy in processing short queries involving out-of-vocabulary terms. We are exploring advancements in open-set tagging capabilities to address this challenge.

---

### Official Review · Reviewer_Kr1V · 2023-10-27

**Soundness:** 4 excellent
**Presentation:** 3 good
**Contribution:** 4 excellent
**Rating:** 6
**Confidence:** 4

**Summary:**

This paper proposes a new Visual Language Pretraining (VLP) framework named as Tag2Text, which introduces image tagging into vision-language models to guide the learning of visual-linguistic features. It utilizes large-scale annotation-free image tags parsed from image-text pairs through text semantic parsing, and demonstrates a foundational image tagging capability with superior zero-shot performance. Tag2Text re-introduces tag guidance into detector-free vision-language models by seamless integrating image tagging, effectively enhancing the performance of both generation-based tasks and alignment-based tasks. The proposed framework is evaluated over a wide range of downstream benchmarks, and achieves state-of-the-art results with similar model sizes and data scales.

**Strengths:**

1.	The idea of the utilization of image tags parsing from large-scale image-text pairs is interesting, efficient and effective from improving the performance of vision-language models.
2.	The framework of Tag2Text employs a multi-task pretraining approach, including Tagging, Generation, and Alignment. These tasks are reasonable and share the same visual features obtained from the image encoder, guaranteeing the efficiency of the framework.
3.	A large number of experimental results over image tagging, image captioning, and image-text retrieval tasks prove the effective of the proposed methods.

**Weaknesses:**

1.	The detail of text semantic parser is not clear. Although it is based on existing work of [Wu et al. 2019], it should make clear how to obtain the corresponding tags in the paper.
2.	In Fig. 2, it is difficult to understand which are users’ input desired tags, since they share the same forms of the recognized image tags.
3.	In the Image-Tag-Text Generation paragraph, the introduction of this task is not clear. According to Fig. 4(c), it seems the text embedding should not be used as input, which is conflicting with the introduction in the corresponding paragraph.
4.	In the experiment on controllability analysis, it is not clear how the threshold of tagging head to control the tagging guidance.
5.	There are some typos in the manuscript.

**Questions:**

Please try to address the weaknesses.

---

> ### Author Response · Authors · 2023-11-13
> **Response to Reviewer Kr1V**
>
> Thanks for your encouraging words and constructive comments.
>
> We carefully address your concerns in specific comments below. And if you have any further concerns, we would be keen to address them as well.
>
>
> ### More details of text parser. ###
>
> > *"The detail of text semantic parser is not clear. Although it is based on existing work of [Wu et al. 2019], it should make clear how to obtain the corresponding tags in the paper. "*
>
> Thank you for the constructive suggestion.
>
> The parser operates based on the dependency tree, a grammatical structure that maps syntactic relationships within a sentence. With a set of established semantic rules is applied to the dependency tree, the parser identifies *'entities (=head + modifier)'* and their *'relationships'* based on the given sentence. For instance, given the sentence "A red alarm clock is on a wooden desk", the parser automatically parse this as: *'head': ['alarm clock', 'desk']*, *'modifier': ['red', 'wooden']*, *'relation': ['on']*.
>
>
> We have included these details and examples in the Section 3.2.
>
>
>
> ### Clarification on 'users specified tags' in Figure 2. ###
>
> > *"In Fig. 2, it is difficult to understand which are users’ input desired tags, since they share the same forms of the recognized image tags. "*
>
> We are sorry for this confusion. In Figure 2, all green underlined tags within *'Tag2Text (User Specified)'* refer to user-specified tags. And we try to clarify Figure 2:
>
> *  Benefits from the powerful tagging capability of Tag2Text, Tag2Text can integrate the recognized image tags to automatically generate comprehensive and detailed captions.
>
>
> *  Tag2Text also allows users to input specified tags to generate corresponding captions. This flexibility, as highlighted by Reviewer Nwx3, offers a way of controlling caption generation through the use of input tags.
>
>
> We have refined the captions of Figure 2 to clearly distinguish them.
>
>
>
> ### Clarification on Image-Tag-Text Generation Task. ###
>
> > *"In the Image-Tag-Text Generation paragraph, the introduction of this task is not clear. According to Fig. 4(c), it seems the text embedding should not be used as input, which is conflicting with the introduction in the corresponding paragraph. "*
>
>
>  Sorry for this confusion. During the Image-Tag-Text Generation training process, the text embeddings are utilized as ground truths to optimize the model via Language Modeling Loss. In the Image text generation comparison of Figure 4, all the text embeddings are utilized as ground truths during training and as model outputs during inference.
>
>
> Thanks for your question, we have clarified these details Section 3.3.
>
>
> ### Clarification on tagging guidance. ###
>
> > *"In the experiment on controllability analysis, it is not clear how the threshold of tagging head to control the tagging guidance. "*
>
>
> During the image tagging inference process, the tagging head outputs logits (ranging from 0 to 1) for each category. These logits are compared to a set threshold to determine the output tags. When the logit exceed this threshold, the corresponding tag category is outputted. Therefore, the tagging guidance can be controlled by adjusting the threshold. For instance, a lower threshold yields more image tags, resulting in higher recall. On the contrary, a higher threshold increases precision.
>
>
> Thank you for the valuable question. We have added above details in Section C of our revised manuscript for better clarity.

---

> > ### Author Response · Authors · 2023-11-23
> > **A kind reminder**
> >
> > Dear Reviewer Kr1V:
> >
> > Thanks again for your encouraging words and constructive comments. The deadline for the public comment is approaching. We are keen to ensure that our revisions and responses align with your expectations and address all your concerns effectively. Please feel free to let us know if you have other questions, we want to solve them for you.
> >
> > Best,
> >
> > Authors

---

### Official Review · Reviewer_gXiT · 2023-10-30

**Soundness:** 3 good
**Presentation:** 2 fair
**Contribution:** 2 fair
**Rating:** 3
**Confidence:** 5

**Summary:**

This work introduces TAG2TEXT, incorporating object tag information into vision-language pre-training. The authors demonstrate its effectiveness across various downstream tasks. In comparison to widely-used object detectors, this method is not only faster but also enriches the model with finer-grained information.

**Strengths:**

- The idea is straightforward and easy to grasp.
- Given the high cost of human labeling and the limited diversity in generating captions for Image Captioning models, such as BLIP, finding new methods for generating comprehensive captions is both challenging and valuable.
- This work offers a potential benefit in the context of person re-identification (REID).
- The motivation behind this approach is also quite appealing.
- Notably, the Tag method diverges from the commonly used Faster R-CNN-based object detectors and demonstrates significantly improved speed.

**Weaknesses:**

- Data and Pre-training Settings: The author of the paper used a 4M setting, which includes training data from COCO (Common Objects in Context) and Visual Genome. NoCaps data is sourced from OpenImages and COCO. Importantly, this work does not incorporate any out-of-distribution data. The success of this approach on COCO-related tasks is attributed to the similarity between the pre-training data and COCO style, and it's noted that many works beyond BLIP face similar challenges.

- TagEval Task: TagEval is mentioned as a task, but it is not considered popular, and its persuasiveness is limited. Models trained on tags are noted to perform well in this case.

- Tag Introduction in Pre-training: The introduction of tags in pre-training is not a novel idea. There are existing works, like OSCAR, that have explored the concept of using tags to improve vision-language pre-training.

**Questions:**

In Figure 1(b), the authors mention 'Actions.' Could you please clarify the source of this particular type of data or tag? Are there any documented instances of failure cases for Tag2Text?

---

> ### Author Response · Authors · 2023-11-13
> **Response to Reviewer gXiT (1/2)**
>
> Thank you for taking the time to read and review our paper.
>
> We carefully address your concerns in specific comments below. And if you have any further concerns, we would be keen to address them as well.
>
> ### Evaluation on Out-Of-Distribution Dataset. ###
> > *"This work does not incorporate any out-of-distribution data. The success of this approach on COCO-related tasks is attributed to the similarity between the pre-training data and COCO style, and it's noted that many works beyond BLIP face similar challenges. "*
>
> We recognize your major concern about evaluating on out-of-distribution datasets. We try to address this from two perspectives.
>
> * **Text Perspective**. In our study, we follow the official NoCaps benchmark [1] and other well-established studies [2-8] as our training and evaluation settings. Notably, NoCaps categorizes benchmarks into in-distribution and **out-of-distribution subsets** based on COCO labels. In Table 5 of our paper, we report these metrics demonstrating that Tag2Text outperforms existing models, particularly in out-of-distribution scenarios. This evidences the ability of Tag2Text to generalize beyond COCO-style datasets in handling a **diverse range of textual contexts**.
>
> * **Image Perspective**. In addition to textual contexts, we shared the same viewpoint on the necessity of including more out-of-distribution images in evaluation benchmarks. Therefore, in Table 1 of our paper, we have made progress for conducting comprehensive evaluation on the out-of-distribution OpenImages benchmark [9], which is a well-established benchmark in image tagging [10-12]. Significantly, Tag2Text achieves superior zero-shot performance even comparable to fully supervised models (**Tag2Text: 83.4 (zero-shot) vs. ML-Decoder 85.5 (full supervised)**), **without including any OpenImages training images in its training stage**. This result demonstrates the effectiveness of Tag2Text across **diverse image domains**.
>
>
> Based on above analysis, we respectfully disagree with the view of *"this work does not incorporate any out-of-distribution data."* Since our experiments includes evaluation on out-of-distribution datasets to proves the effectiveness of our proposed methods, which is appreciated by Reviewer Kr1V and Nwx3.
>
>
> ### Novelty Concerns in Tag Introduction. ###
>
> > *"Tag Introduction in Pre-training: The concept of using tags in pre-training is not new, with examples like OSCAR in the field. "*
>
>
> We acknowledge this point and have tried our best to articulate our novel contributions, especially in comparison to works like OSCAR, in the Introduction of our paper:
>
>
> * **Unidirectional vs. Bidirectional Improvements.** OSCAR utilizes tags to enhance vision-language models in a unidirectional manner. For getting image tags, OSCAR directly utilizes an off-the-shelf object detector (Faster RCNN).
> In contrast, Tag2Text offers **bidirectional enhancements on both image tagging and vision-language models**. For image tagging, **Tag2Text pioneer utilizing large-scale image-text pairs instead of manual annotations, achieving superior zero-shot ability.**
>
>
> * **Object Detector vs. Image Tagging**. Although using tags based on the object detector is useful, the trend in recent studies [4-8,13-14] is **shifting towards detector-free vision-language models**. Since the frozen object detector restricts the model's capacity and is time-consuming. Tag2Text **re-introduces the valuable tag guidance** into detector-free models via image tagging. This approach enables data scaling, and is both efficient and effective in improving VL tasks.
>
>
> ### Clarifications on 'Actions' in Figure 1(b) ###
>
> > *"In Figure 1(b), the authors mention 'Actions.' Could you please clarify the source of this particular type of data or tag?  "*
>
> We are sorry for this confusion.
>
> In Section 3.2, we detail the source of image tags. Specifically, we parse 4 million image-text pairs and select the most frequently occurring tag categories as our label system, including objects, scenes, attributes, actions. The action categories  (e.g., 'fly', 'stand', 'climb') are obtained from the *'relationship'* items after text parsing. We also provide more tag category details in Appendix B, including tag category statistics in Table 7, an illustration in Figure 8, and the overlapping with other public datasets in Table 8 and Table 9.
>
> We have added an example in Section 3.2 of our revised manuscript.

---

> ### Author Response · Authors · 2023-11-13
> **Response to Reviewer gXiT (2/2)**
>
> ### Failure Cases of Tag2Text ###
> > *"Are there documented instances of failure cases for Tag2Text?"*
>
> * **Hallucinatory Captions**. Tag2Text benefits from its powerful tagging capabilities. As depicted in Figure 6, there is a strong correlation between captioning performance and tagging guidance performance. In practical applications, we observe that incorrect user-provided tags may lead to hallucinatory captions.
>
> * **Small Objects**. In addition, evaluating a tagging model capable on 3,429 categories is also challenging. Our quantitative comparison and visual validations reveal that Tag2Text efficiently recognizes common objects and scenes, yet struggles in small objects (e.g., spoon or baseball). Our empirical experiments indicate that increasing the resolution during fine-tuning significantly improves performance on these small objects.
>
> Thanks for your constructive question, we have added the Section G of "Limitation" in our revised manuscript.
>
>
> ### Reference ###
>
> [1] Nocaps: Novel object captioning at scale, in CVPR 2019.
>
> [2] Oscar: Object-Semantics Aligned Pre-training for Vision-Language Tasks, in ECCV 2020.
>
> [3] VinVL: Revisiting Visual Representations in Vision-Language Models, in CVPR 2021.
>
> [4] BLIP: Bootstrapping Language-Image Pre-training for Unified Vision-Language Understanding and Generation, in ICML 2022.
>
> [5] Multi-Grained Vision Language Pre-Training: Aligning Texts with Visual Concepts, in ICML 2022.
>
> [6] Injecting Semantic Concepts Into End-to-End Image Captioning, in CVPR 2022.
>
> [7] GIT: A Generative Image-to-text Transformer for Vision and Language, in Arxiv 2022.
>
> [8] BLIP-2: Bootstrapping Language-Image Pre-training with Frozen Image Encoders and Large Language Models, in ICML 2023.
>
> [9] The open images dataset v4. in IJCV 2020.
>
> [10] Asymmetric Loss for Multi-Label Classification, in ICCV 2021.
>
> [11] Simple and Robust Loss Design for Multi-Label Learning with Missing Labels, in Arxiv 2021
>
> [12] ML-Decoder: Scalable and Versatile Classification Head, in WACV 2023.
>
> [13] Align before Fuse: Vision and Language Representation Learning with Momentum Distillation, in NeurIPS 2021.
>
> [14] An Empirical Study of Training End-to-End Vision-and-Language Transformers, in CVPR 2022

---

> ### Author Response · Authors · 2023-11-18
> **Addition Zero-Shot Evaluations**
>
> # Addition Zero-Shot Evaluations
>
> Regarding to your major concern of "Evaluation on Out of Distribution Dataset", we have responded in "Response to Reviewer gXiT (1/2)", that **we have conducted zero-shot evaluations on NoCpas and OpenImages**.
>
> Despite the fact that we have conducted sufficient experiments, to further address your concern, we conducted addition zero-shot evaluations on **NUS-WIDE** [1], a well-established tagging benchmark including 81 categories. Similarly, **all images in NUS-WIDE are out-of-distribution data**, since Tag2Text did not utilize any NUS-WDIE training images during its training process. The results are presented in the table below and have been included in Section E of the revised manuscript.
>
>
> Notably, **Tag2Text also demonstrates superior zero-shot performance**, exceeding both CLIP and BLIP, while utilizing much less training data. We find it challenging to find another zero-shot benchmark for image captioning besides NoCpas. However, considering Tag2Text benefits from its tagging ability, we contend these results demonstrate the powerful generalization ability of Tag2Text.
>
>
> | Zero-Shot Nus-Wide Comparison | Pretrain Images | F1   | Precision | Recall |
> |-------------------------------|-----------------|------|-----------|--------|
> | CLIP                          | 400M            | 46.0 | 54.0      | 40.1   |
> | BLIP                          | 129M            | 45.0 | 54.0      | 38.6   |
> | Tag2Text                      | 14M             | **46.4** | **54.7**      | **40.3**   |
>
>
> Please feel free to share if you have further concerns, we would be keen to address them as well.
>
> [1] NUS-WIDE: a real-world web image database from National University of Singapore, ACM MM 2009.

---

> > ### Author Response · Authors · 2023-11-23
> > **A kind reminder**
> >
> > Dear Reviewer gXiT:
> >
> > Thanks again for your detailed comments and helpful feedback. The deadline for the public comment is approaching. We are keen to ensure that our revisions and responses align with your expectations and address all your concerns effectively. Please feel free to let us know if you have other questions, we want to solve them for you.
> >
> > Best,
> >
> > Authors

---

### Official Review · Reviewer_pdv8 · 2023-10-31

**Soundness:** 4 excellent
**Presentation:** 4 excellent
**Contribution:** 2 fair
**Rating:** 6
**Confidence:** 5

**Summary:**

This paper describes a straightforward approach to improve Vision-Language models by using tag information extracted from image captions. By utilising standard techniques and losses they show very decent results on tagging, captioning and retrieval tasks

**After rebuttal**: I appreciate the authors' rebuttal. I think the original comments I made are still relevant. The paper definitely makes a decent contribution and I wouldn't mind seeing it published at ICLR. However I'm not sure I can increase my score to 8.

**Strengths:**

1. The paper is well written and easy to follow.
2. The method is simple and is shown to work well. The idea of using image tags to aid VL pre-training makes a lot of sense.
3. The results are not always SOTA but are very good. Results on multiple tasks/datasets are provided. I believe the results are sufficient to show that the main idea behind the paper works as well as expected.

**Weaknesses:**

I think the main problem with the paper is that all of its components have been proposed before so the paper looks more like a re-implementation of known ideas with more recent architectures and pipelines which is of course expected to work better. Specifically the main idea of using tags to aid VL pre-training appears in many works including OSCAR or more recently in DiHT (Filtering, Distillation, and Hard Negatives for Vision-Language Pre-Training) while other losses used like I2T are very commonly used in most works in VL pre-training. But the method for sure can serve as strong baseline.
Somewhat less important concern: as most experiments are on COCO/Flick which are datasets very close to the ones used for training I am wondering whether the authors could carry out an experiment on out-of-domain data .

**Questions:**

No specific questions

---

> ### Author Response · Authors · 2023-11-13
> **Response to Reviewer pdv8**
>
> We sincerely appreciate your encouraging feedback and constructive comments. Our point-to-point responses to your comments are given below.
>
>
> ###  Introduction of Tag Guidance. ###
>
> > *"The main idea of using tags to aid VL pre-training appears in pervious works such as OSCAR. But the method for sure can serve as strong baseline."*
>
> We acknowledge this point and have tried our best to articulate our novel contributions, especially in comparison to works like OSCAR, detailed in the Introduction our paper:
>
>
> * **Unidirectional vs. Bidirectional Improvements.** OSCAR utilizes tags to enhance vision-language models in a unidirectional manner. For getting image tags, OSCAR directly utilizes an off-the-shelf object detector (Faster RCNN).
> In contrast, Tag2Text offers **bidirectional enhancements on both image tagging and vision-language models**. For image tagging, **Tag2Text pioneer utilizing large-scale image-text pairs instead of manual annotations, achieving superior zero-shot ability.**
>
>
> * **Object Detector vs. Image Tagging**. Although using tags based on the object detector is useful, the trend in recent studies [1-7] is **shifting towards detector-free vision-language models**. Since the frozen object detector restricts the model's capacity and is time-consuming. Tag2Text **re-introduces the valuable tag guidance** into detector-free models via image tagging. This approach enables data scaling, and is both efficient and effective in improving VL tasks.
>
> ### Evaluation on out-of-distribution dataset. ###
>
> > *"Somewhat less important concern: As most experiments are on COCO/Flick which are datasets very close to the ones used for training I am wondering whether the authors could carry out an experiment on out-of-domain data."*
>
> Our experiments include comprehensive evaluation on out-of-distribution datasets:
>
> * **Image Captioning.** Following other well-established studies [3-9], we perform zero-shot evaluation on the NoCaps benchmark [10]. Notably, NoCaps categorizes benchmarks into in-distribution and **out-of-distribution subsets** based on COCO labels. In Table 5 of our paper, we report these metrics demonstrating that Tag2Text outperforms existing models, particularly in out-of-distribution scenarios. This evidences the ability of Tag2Text to generalize beyond COCO-style datasets in handling a **diverse range of textual contexts**.
>
>
> * **Image Tagging.** We further extend our evaluation on the out-of-distribution OpenImages benchmark [11], which is a well-established benchmark in image tagging [12-14]. Significantly, Tag2Text achieves **superior zero-shot performance** even comparable to fully supervised models, **without including any OpenImages training images in its training stage**. This result demonstrates the effectiveness of Tag2Text across **diverse image domains**.
>
> Furthermore, we have also conducted an addition zero-shot evaluations on NUS-WIDE [15] in Section E of our revised manuscript.
>
> We hope our responses satisfactorily address your concerns. And if you have any further concerns, we would be keen to address them as well.
>
>
>
> [1] Align before Fuse: Vision and Language Representation Learning with Momentum Distillation, in NeurIPS 2021.
>
> [2] An Empirical Study of Training End-to-End Vision-and-Language Transformers, in CVPR 2022
>
>
> [3] BLIP: Bootstrapping Language-Image Pre-training for Unified Vision-Language Understanding and Generation, in ICML 2022.
>
> [4] Multi-Grained Vision Language Pre-Training: Aligning Texts with Visual Concepts, in ICML 2022.
>
> [5] Injecting Semantic Concepts Into End-to-End Image Captioning, in CVPR 2022.
>
> [6] GIT: A Generative Image-to-text Transformer for Vision and Language, in Arxiv 2022.
>
> [7] BLIP-2: Bootstrapping Language-Image Pre-training with Frozen Image Encoders and Large Language Models, in ICML 2023.
>
> [8] Oscar: Object-Semantics Aligned Pre-training for Vision-Language Tasks, in ECCV 2020.
>
> [9] VinVL: Revisiting Visual Representations in Vision-Language Models, in CVPR 2021.
>
> [10] Nocaps: Novel object captioning at scale, in CVPR 2019.
>
> [11] The open images dataset v4. in IJCV 2020.
>
> [12] Asymmetric Loss for Multi-Label Classification, in ICCV 2021.
>
> [13] Simple and Robust Loss Design for Multi-Label Learning with Missing Labels, in Arxiv 2021
>
> [14] ML-Decoder: Scalable and Versatile Classification Head, in WACV 2023.
>
> [15] NUS-WIDE: a real-world web image database from National University of Singapore, ACM MM 2009.

---

> ### Author Response · Authors · 2023-11-15
> **Addition Comparison with DiHT**
>
> # Comparison with DiHT
>
> We also notice your feedback regarding DiHT [1]. We further clarify the differences between Tag2Text and DiHT.
>
> ## Additional Image Captioning
>
> A significant difference is that DiHT is a alignment model as CLIP. Conversely, Tag2Text provides comprehensive and and detailed image captions, benefits from its powerful tagging capabilities.
>
> ## Advanced Image Tagging
>
> In the table below, we further compare the zero shot tagging performance of DiHT on OpenImages. Notably, despite DiHT's improvements over CLIP, it falls short of Tag2Text, even though DiHT utilizes a significantly larger training image dataset. We attribute to their different alignment paradigms:
>
> * **Gobal Image Features with Text**. Although DiHT filters image text pairs through text parsing, it relies on the alignment of global image features with text via dot product interaction, similar to CLIP.
>
>
> * **Sptial Image Features with Tag**.
> Tag2Text introduces a more fine-grained alignment of image spatial features with tags (parsed from texts) through an efficient recognition decoder. This approach is particularly effective for image tagging, since the tags often correspond to multiple image regions and reside at the token level within the text.
>
>
> | Zero-Shot OpenImages Comparison | Training Images | F1   | Precision | Recall |
> |---------------------------------|-----------------|------|-----------|--------|
> | CLIP                            | 400M            | 63.0 | 77.9      | 52.9   |
> | DiHT                            | 438M            | 66.3 | 77.0      | 65.3   |
> | Tag2Text                        | 14M             | **72.7** | **80.1**      | **66.6**   |
>
>
> We have incorporated these insights into our revised manuscript to further highlight its contributions. Specifically, we have emphasized the fine-grained tagging alignment in Section 3.3 and added the comparison results of DiHT in Table 1.
>
> We appreciate your valuable feedback which inspires us a lot.
>
>
> [1] Filtering, Distillation, and Hard Negatives for Vision-Language Pre-Training, in CVPR 2023.

---

> > ### Author Response · Authors · 2023-11-23
> > **A kind reminder**
> >
> > Dear Reviewer pdv8:
> >
> > Thanks again for your encouraging feedback and constructive comments. The deadline for the public comment is approaching. We are keen to ensure that our revisions and responses align with your expectations and address all your concerns effectively. Please feel free to let us know if you have other questions, we want to solve them for you.
> >
> > Best,
> >
> > Authors

---

### Official Review · Reviewer_N6Aw · 2023-11-01

**Soundness:** 3 good
**Presentation:** 3 good
**Contribution:** 3 good
**Rating:** 5
**Confidence:** 3

**Summary:**

This paper introduces a novel framework for vision-language pre-training. The core insight here is that extracting tags from text paired with images offers robust semantic guidance for vision-language pre-training, and these tags can be easily mined from an existing pipeline. Leveraging these mined tags, the paper presents a multi-task learning framework designed for vision-language pre-training. This framework concurrently handles image captioning, image tagging (multi-label classification), and Image-Text Matching. The paper conducts an extensive array of experiments, with the results demonstrating the significant impact of image tagging in this pre-trained framework.

**Strengths:**

1. Including image tagging task to vision language pretraining seems to be promising, which particularly benefit image captioning.
2. The proposed design that combines multi-tasks for vision language pretraining is interesting.
2. The experimental results are promising.
3. The paper is well-written in general and easy to comprehend.

**Weaknesses:**

1. This paper discusses VL pretraining methods that are either based on 1) object detection or 2) image tagging (the proposed method) and argues that image tagging is faster and introduces significantly fewer parameters. However, this may not be a compelling motivation for choosing image tagging, as object detectors are fixed (with no additional learnable parameters) and only need to be executed once before training, incurring marginal computational cost compared to the training phase.

2. It would be beneficial for this paper to incorporate specific mathematical formulations to provide a more comprehensive description and discussion of the fundamental problems that require resolution.

3. There is room for improvement in the typesetting.

4. Table 1 appears to be disorganized, making it unclear which numbers to focus on and compare. It is advisable to separate results for different tasks into distinct tables.

5. The "SOTA" comparison for image tagging (multi-label classification) seems to omit a substantial portion of recent works. As a result, the reviewer maintains a skeptical stance concerning the associated claims and conclusions.

6. Metrics such as Precision, Recall, and F1 score, which are commonly used for image tagging and multi-label classification, are notably absent from the results.

7. The construction of the tag category system appears to involve human annotation in the process (Section 3.2), which contradicts the earlier claim of being an "automatic" approach (Section 1).

8. Conducting an ablation study on the choice of vocabulary set (tag set) size to be mined could offer valuable insights into the proposed method.

**Questions:**

1. Could the author give a short explanation of the meaning of title "Tag2Text"?
2. Performing a multi-label classification with 3,429 categories is non-trivial. The reviewer wonder if the authors come across any difficulties? And what the performance of the learned classifiers on those category (for example, on a validation set of VL pretraining)?
3. Have the authors ablate VL pretraining framework with a single task, for example, Image-Text alignment only based VL, Image-tagging only based VL, Image captioning only based VL?

---

> ### Author Response · Authors · 2023-11-13
> **Response to Reviewer N6Aw (1/2)**
>
> We sincerely appreciate your detailed comments and helpful feedback on our work.
>
> We carefully address your concerns in specific comments below. And if you have any further concerns, we would be keen to address them as well.
>
> ### Object Detector vs. Image Tagging. ###
> > *"This paper discusses VL pretraining methods that are either based on 1) object detection or 2) image tagging (the proposed method) and argues that image tagging is faster and introduces significantly fewer parameters. However, this may not be a compelling motivation for choosing image tagging, as object detectors are fixed and only need to be executed once before training, incurring marginal computational cost compared to the training phase."*
>
> Thanks for your insightful question. We try to address this from two perspectives.
>
> * **Capacity Limitation**. Despite a frozen object detector reduces computational costs during training, it **lacks the ability for bidirectional optimization in an end-to-end manner**. Consequently, the inherent limitations in a frozen object detector can restrict the capabilities of vision-language (VL) models, as highlighted in [1-3]. More recent studies [1-6] have shifted towards detector-free VL models, resulting in the discarding of valuable tags.
>
>     To this end, we re-introduce tagging guidance into detector-free models. Our proposed image tagging utilizes large scale image tags in accordance with image-text pairs, **ensuring the capabilities of both VL tasks and image tagging**.
> * **Inference Efficiency**.  Regarding inference efficiency, object detectors are typically more time-consuming compared to image tagging (e.g., **Object Detector: 153ms** vs. **Image Tagging: 40ms**).
>
> As a result, our image tagging enables data scaling, and is both effective and efficient in improving VL tasks.
>
> ### Mathematical Formulations. ###
> > *" It would be beneficial for this paper to incorporate specific mathematical formulations to provide a more comprehensive description and discussion of the fundamental problems that require resolution."*
>
> Thanks for your valuable suggestion.
>
> We have provided mathematical formulations to detail different pre-training tasks in Appendix A.2.
>
> ### SOTA Comparison for Image Tagging. ###
> > *" The "SOTA" comparison for image tagging (multi-label classification) seems to omit a substantial portion of recent works. As a result, the reviewer maintains a skeptical stance concerning the associated claims and conclusions."*
>
> Thanks for your constructive question.
>
> To assess the tagging capability of Tag2Text, recognizing 3,429 categories, we employed multiple benchmarks and selected **SOTA models capable on these benchmarks**. Our comparison includes both tagging models (e.g., ML Decoder [7], MKT [8]) and VL models (e.g., CLIP [9], BLIP [4], BLIP-2 [7]). Notably, ML-Decoder continues to demonstrate exceptional performance on the OpenImages LeaderBoard [10], serving as a **powerful fully supervised benchmark** for our analysis.
>
> As detailed in Table 1 of our paper, Tag2Text achieves superior zero-shot performance on OpenImages, without including any OpenImages training images in its training stage. Significantly, this performance is comparable to fully supervised models of ML-Decoder (**Tag2Text: 83.4 (zero-shot) vs. ML-Decoder 85.5 (full supervised)**). We contend these results highlights the robust tagging capabilities of Tag2Text.
>
> ### Evaluation Metrics for Image Tagging. ###
> > *" Metrics such as Precision, Recall, and F1 score, which are commonly used for image tagging and multi-label classification, are notably absent from the results."*
>
> Thanks for your question. As detailed in Table 1 of our paper, we have indeed included a variety of evaluation metrics, encompassing mAP, Precision, and Recall, to comprehensively assess our model's performance.
>
> * **mAP**. We utilize mAP as the primary metric when comparing with other tagging models, due to its well-established status in the field of image tagging.
>
> * **Precision and Recall**. For comparison with vision-language models, where direct calculation of mAP is challenging, we employ Precision and Recall as our main metrics.
>
> In response to your feedback, the additional F1 score comparisons are presented below. Notably, Tag2Text demonstrates significant enhancements across various datasets. These results have been included in Table 2 of our revised manuscript.
>
>
>
> | F1 Comparison | Training Images | Evaluation Paradigm | TagEval | OpenImages | COCO |
> |---------------|-----------------|---------------------|---------|------------|------|
> | CLIP| 400M  | Alignment | 63.4| 63.0| 48.2 |
> | BLIP| 129M  | Alignment | 65.7| 64.8| 54.3 |
> | BLIP| 129M  | Captioning| 58.6| 56.6| 55.7 |
> | BLIP-2 | 129M  | Captioning| 58.2| 58.1| 59.1 |
> | Tag2Text| 14M   | Captioning| 65.9| 62.7| 62.7 |
> | Tag2Text| 14M   | Tagging   | **78.6**| **72.7**| **71.5** |

---

> ### Author Response · Authors · 2023-11-13
> **Response to Reviewer N6Aw (2/2)**
>
> ### Clarification on Tag Category System Construction ###
> > *" The construction of the tag category system appears to involve human annotation in the process (Section 3.2), which contradicts the earlier claim of being an "automatic" approach (Section 1)."*
>
> We are sorry for this confusion.
>
> Extracting large-scale tags from texts through automatic text parsing is entirely automated with no manual annotation, as stated in Section 1: *"The pre-training image tags are obtained through automatically text semantic parsing"*
>
> In Section 3.2, in order to build a more comprehensive label system, we filter the tag categories with a minimal amount of manual effort. For example, we merge synonyms such as 'person' and 'human'. We have added this example in Section 3.2 of our revised manuscript.
>
> ### Impact on Vocabulary Set Size ###
> > *"Conducting an ablation study on the choice of vocabulary set (tag set) size to be mined could offer valuable insights into the proposed method.."*
>
> We appreciate this constructive suggestion!
>
> In response, we have expanded the vocabulary set from 3,429 to 4,585 categories and compared the performances, as presented in the table below. Notably, the tagging performance decrease with the larger vocabulary set. We attribute to two possible reasons:
>
> * The increased complexity in training with more categories.
>
> * The additional categories leading to more noise, as they lack sufficient training data, thereby impacting the model's efficiency.
>
> These results and analyses have been incorporated in Section B of our revised manuscript.
>
> | Vocabulary Set Size|TagEval|OpenImages|
> |-|-|-|
> | 3,429 | 81.7| 84.1|
> | 4,585 | 80.3| 83.1|
>
> ### Clarification on Model Name "Tag2Text" ###
> > *"Could the author give a short explanation of the meaning of title "Tag2Text"?."*
>
> Sorry for the confusion. The name "Tag2Text" is chosen to vividly represent the core functionality of our model, which refers to the integration of image tags as guiding elements into text generation. Benefits from its powerful tagging capabilities, Tag2Text is able to generate more comprehensive and and detailed text descriptions, as illustrated in Figure 2 of our paper.
>
> In addition, Tag2Text also allows users to input specified tags to generate corresponding captions. This flexibility, as highlighted by Reviewer Nwx3, offers a way of controlling caption generation through the use of input tags.
>
> ### Image Tagging with 3,429 Categories ###
> > *"Performing a multi-label classification with 3,429 categories is non-trivial. The reviewer wonder if the authors come across any difficulties? And what the performance of the learned classifiers on those category?"*
>
> Thank you for raising this important point!
>
> We share the same viewpoint that performing a tagging model with 3,429 categories presents significant challenges, primarily due to the scarcity of training data for such a large number of categories. To address this, we pioneer the utilization of large-scale image-text pairs instead of manual annotations. This approach, combined with our multi-task and pretrain-finetune manners, are proven effective for image tagging as detailed in Table 5 of our paper.
>
>
> Evaluating a tagging model on such a wide category range is also challenging. Our quantitative comparison and visual validations reveal that Tag2Text efficiently recognizes common objects and scenes, yet struggles in small objects (e.g., spoon or baseball). Our empirical experiments indicates that increasing the resolution during fine-tuning significantly improves performance on these small objects.
>
> Thanks for your constructive question, we have added the Section G of "Limitation" in our revised manuscript.
>
>
> ### Ablation Study with Single Task ###
> > *"Have the authors ablate VL pretraining framework with a single task, for example, Image-Text alignment only based VL, Image-tagging only based VL, Image captioning only based VL?."*
>
> Thanks for your insightful question.
>
> In our paper, we have reported the results with only either VL tasks or tagging task. We summarize them in the table below. The table clearly demonstrates the effectiveness of our bidirectional enhancements on both image tagging and VL tasks.
>
> |Pre-Training Objectives|Source of our Paper|Tagging|Captioning| Retrieval |
> |-|-|-|-|-|
> |Tagging|Table 5 Row 5|74.7|-|-|
> |Captioning + Alignment |Table 4 Row 2 |-|123.3|74.4|
> |Tagging + Captioning + Alignment|Table 5 Row 6 / Table 4 Row 3|76.4 / 82.9(+ft)|124.6|74.9|

---

> > ### Author Response · Authors · 2023-11-13
> > **Reference**
> >
> > [1] Align before Fuse: Vision and Language Representation Learning with Momentum Distillation, in NeurIPS 2021.
> >
> > [2] An Empirical Study of Training End-to-End Vision-and-Language Transformers, in CVPR 2022
> >
> > [3] Multi-Grained Vision Language Pre-Training: Aligning Texts with Visual Concepts, in ICML 2022.
> >
> > [4] BLIP: Bootstrapping Language-Image Pre-training for Unified Vision-Language Understanding and Generation, in ICML 2022.
> >
> > [5] Injecting Semantic Concepts Into End-to-End Image Captioning, in CVPR 2022.
> >
> > [6] BLIP-2: Bootstrapping Language-Image Pre-training with Frozen Image Encoders and Large Language Models, in ICML 2023.
> >
> > [7] ML-Decoder: Scalable and Versatile Classification Head, in WACV 2023.
> >
> > [8] Open-Vocabulary Multi-Label Classification via Multi-Modal Knowledge Transfer, in AAAI 2023.
> >
> > [9] Learning Transferable Visual Models From Natural Language Supervision, in ICML 2021.
> >
> > [10] [LeaderBoard: Multi-Label Classification on OpenImages-v6](https://paperswithcode.com/sota/multi-label-classification-on-openimages-v6)

---

> ### Author Response · Authors · 2023-11-23
> **A kind reminder**
>
> Dear Reviewer N6Aw:
>
> Thanks again for your detailed comments and helpful feedback. The deadline for the public comment is approaching. We are keen to ensure that our revisions and responses align with your expectations and address all your concerns effectively. Please feel free to let us know if you have other questions, we want to solve them for you.
>
> Best,
>
> Authors

---

### Author Response · Authors · 2023-11-20
**Further Paper Revision**

Thanks to all the reviewers for the valuable suggestions. We have carefully revised our manuscript, with changes marked in blue for clarity. The major modifications include:

* Provide F1 metric for tagging evaluation in Table 2.
* Include compared models of DiHT [1] in Table 2.
* Conduct addition zero-shot evaluations on NUS-WIDE [2] in Section E.
* Analyze fine-grained alignment of Tag2Text compared to CLIP [3] in Section 3.3.
* Evaluate the impact on vocabulary set size in Section C.
* Discuss limitations of Tag2Text in Section G.
* Fix typos and enhance more details (user-specified tags, mathematical formulas, text parser, tagging threshold, zero-shot illustrations of Table 3 and Table 5).

Please feel free to share if you have further concerns, we would be keen to address them as well.

[1] Filtering, Distillation, and Hard Negatives for Vision-Language Pre-Training, in CVPR 2023.

[2] NUS-WIDE: a real-world web image database from National University of Singapore, ACM MM 2009.

[3] Learning Transferable Visual Models From Natural Language Supervision, in ICML 2021.

---

### Author Response · Authors · 2023-11-22
**A kind reminder**

Dear reviewers:

We have submitted the reply a few days ago. Now the deadline for public comment is approaching. We are keen to ensure that our revisions and responses align with your expectations and address all your concerns effectively. Please feel free to let us know if you have other questions.

Best,

Authors

---

### Meta-Review · Area_Chair_fgvY · 2023-12-06

**Metareview:**

This paper proposes a novel framework for vision-language pre-training. It extracts  tags   parsed from its paired text to provide more  semantic guidance in the pre-training phase. Then the paper presents a multi-task learning framework designed for vision-language pre-training. Experimental results show that the proposed framework can well handle several tasks, such as image captioning, image tagging, and Image-Text Matching.

Almost all reviewers agree with the contributions of the proposed method, and its sufficient experiments.  But the authors should also well discuss the novelty issue that X-VLM[1] already generates both bounding box information and object tags via a detector, and includes all experimental results into the revision. Since most reviewers agree with accept this work, we decide to accept it.

**Justification For Why Not Higher Score:**

One reviewer who gives a score of 3 argues the novelty of this work, since  X-VLM[1] already generates both bounding box information and object tags via a detector. The authors claim this work is a detector-free vision-language learning and is more efficient than previous one. Such a claim is accepted by other reviewers.

**Justification For Why Not Lower Score:**

1) Almost all reviewers agree with the contributions of the proposed method, and its sufficient experiments.

2) One reviewer who gives a score of 3 argues the novelty of this work, since  X-VLM[1] already generates both bounding box information and object tags via a detector. The authors claim this work is a detector-free vision-language learning and is more efficient than previous one. Such a claim is accepted by other reviewers.

3) one reviewer who gives a score of 6 said he would like to increase the score from 6 to 8. So the average score will be 6 which is a positive score.

---

### Decision · Program_Chairs · 2024-01-16

Accept (poster)